# Magnetohydrodynamics Simulation of the Nonlinear Behavior of Single Rising Bubbles in Liquid Metals in the Presence of a Horizontal Magnetic Field

**Marino Corrado** [1,2] **and Yohei Sato** [2,*]

1    Department of Mechanical and Process Engineering, ETH Zurich, 8092 Zurich, Switzerland
2    Division Scientific Computing, Theory and Data, Paul Scherrer Institute, 5232 Villigen PSI, Switzerland
*    Correspondence: yohei.sato@psi.ch; Tel.: +41-56-310-26-66

**Abstract:** Rising bubbles in liquid metals in the presence of magnetic fields is an important phenomenon in many engineering processes. The nonlinear behavior of the terminal rise velocities of the bubbles as a function of increasing field strength has been observed experimentally, but it remains poorly understood. We offer an explanation of the phenomenon through numerical calculations. A single rising bubble in stagnant liquid metal in the presence of an applied horizontal magnetic field is simulated. The observed nonlinear behavior is successfully reproduced; the terminal velocity increases with the increase in the magnetic field strength in the lower magnetic field regions but decreases in higher regions. It is shown that, in the lower region, the increase in the average bubble rise velocity results from the suppression of the fluctuations in the bubble trajectory in the vertical plane perpendicular to the magnetic field, as a consequence of the Lorentz force resulting from the component of induced electric current due to the magnetic field, which (approximately) acts in the opposite direction to that of the flow velocity. For higher magnetic field strengths, the Lorentz force induces a broadened wake in the vertical plane parallel to the applied magnetic field, resulting in a decrease in the rise velocity.

**Keywords:** CFD; MHD; rising bubble; Lorentz force

## 1. Introduction

Recently, the study of bubbly flows in liquid metal in the presence of an external magnetic field has been receiving increasing attention, since the concept has the potential to be used in different engineering processes, such as liquid-metal stirring, purification and casting, by controlling the bubble motion, and hence the mixing capability, using an applied magnetic field. Apart from the metallurgical engineering applications, such a flow situation could also be a feature of nuclear fusion reactors, in which bubbles are injected into the liquid-metal coolant to enhance the efficiency of the heat transfer processes. Due to the opaqueness of the liquid metal, it is impossible to observe the bubbles directly using optical measurement techniques, and thus, indirect non-optical measures, such as local conductivity probes [1–4], Ultrasound Doppler Velocimetry (UDV) [5–8], X-ray radiography [9–11] and neutron radiography [4,12–15], have been employed to "observe" bubble dynamics in the presence of a magnetic field in many related experiments. In parallel to the measurements, numerical Magneto-Hydro-Dynamics (MHD) simulations have been widely employed to better understand bubble dynamics [16–19], since with this approach, the distribution of all the variables related to the flow and electric field can, in principle, be obtained.

In the early stages of making quantitative measurements, Mori, et al. [1] performed a series of experiments involving single nitrogen bubbles rising in quiescent liquid mercury under the influence of a horizontal, uniform and static magnetic field (HMF). The rise velocities and the aspect ratios of the bubbles were measured using the electrical triple

probe method, by which the velocity was calculated from the difference between the bubble release time from the (bottom) injection nozzle and the bubble arrival time at the probe positioned higher in the tank. Thus, the acceleration of the bubble was not explicitly considered, which may have resulted in a lower estimate of the bubble velocity than the actual value. The direct influence of the contact between the probe and the rising bubble is also unclear, and unfortunately, the uncertainty (variance) in the measurement was not reported in the paper. However, it was observed that the rise velocity exhibited nonlinear behavior in the cases of smaller bubbles ($Eo < 2$), with the rise velocity increasing with increases in the magnetic field strength (in the range of $0 < B < 1$ T) but then decreasing with further increases in the magnetic field, in the range of $B > 1.5$ T. Here, the definition of the Eötvös number $Eo$ is defined in the Nomenclature section. Such nonlinear behavior was not observed for larger bubbles ($Eo > 4$), with the rise velocities in these cases decreasing monotonically with increases in the magnetic field. In the current paper, we focus on this nonlinearity and propose an explanation of it.

Experiments involving single bubbles rising in a liquid metal in a vertical (i.e., the same direction as the gravity), uniform and static magnetic field (VMF) were performed by Zhang, et al. [5] at the Helmholtz-Zentrum Dresden-Rossendorf Laboratory. Bubble and liquid velocities were measured using UDV [20], a technique which enables the bubble rise velocity to be measured at arbitrary elevations. The results showed that the rise velocity of small bubbles ($Eo \leq 2.5$) decreases with increases in the applied magnetic field for $B \leq 0.3$ T, whereas the opposite is true in the case of larger bubbles ($Eo \geq 3.4$). A single rising bubble of $Eo = 5$ in a liquid metal under a VMF was numerically studied by Shibasaki, et al. [16] using an MHD simulation, and the results captured the nonlinear behavior of the terminal rise velocity. Schwarz and Fröhlich [21] also performed MHD simulations for single rising bubbles in a liquid metal under a VMF, for which two assumptions were introduced; namely, a no-slip condition was imposed at the bubble surface, and the bubble shape was assumed to be axisymmetric. The feasibility of these assumptions remains an open question, but the results indeed show an increase in the rise velocity with increases in the magnetic field for larger bubbles. Zhang and Ni [22] at the University of the Chinese Academy of Sciences also performed MHD simulations subject to a VMF. Their results showed that larger bubbles ($Eo \geq 2.2$) attain maximum terminal velocity with increases in the magnetic field.

Further MHD simulations for single rising bubbles under HMF conditions were performed by Jin, et al. [17]. The simulations were related to argon bubbles ($1 < Eo < 6$) rising in GaInSn for $0 \leq B < 0.5$ T. In the same year, Zhang, et al. [18] also published simulations of single rising bubbles in an HMF, with the ranges in this case being $2.2 \leq Eo \leq 4.9$ and $0 \leq N \leq 24$, where $N$ is the Stuart number. In both works [17,18], the terminal rise velocity was predicted to monotonically decrease with increases in the magnetic field, i.e., the nonlinear behavior measured by Mori, et al. [1] for a VMF was not observed in the case of an HMF.

Measurements of the rise velocities of single argon bubbles in GaInSn under an HMF were also performed by Wang, et al. [7] using the UDV technique. The experiments covered the ranges of $1 < Eo < 4$ and $0 \leq B < 2$ T. They also observed the nonlinear behavior of the terminal rise velocity, which was previously measured by Mori, et al. [1] for nitrogen bubbles in mercury. In the same year as the publication of Wang, et al. [7], Strumpf [8] reported results from a similar experiment, namely argon bubbles in GaInSn under an HMF using UDV for $1 < Eo < 5$ and $0 \leq B < 1$ T. Strumpf changed the intensity of the magnetic field gradually and thereby observed the influence of the magnetic field intensity on the terminal rise velocity. The results showed that the maximum terminal rise velocity appears at $N/C_d \approx 1$, where $N$ is the Stuart number and $C_d$ is the drag coefficient of the bubble, although no physical explanation was given for the phenomenon.

Very recently, Zhang, et al. [19] presented MHD simulations for both HMF and VMF orientations to better understand the influence of the applied magnetic field on the trajectories of the rising bubbles. The results showed that the spiral motion of a bubble observed

for a zero magnetic field changed to a purely zig-zag motion under the influence of an HMF. To understand the mechanism, they analyzed the evolution of the wake vortices and the forces to which they were subjected. According to their simulation results, oscillations in the bubble trajectory are suppressed in the direction of the applied magnetic field vector.

A comprehensive list of the experiments and simulations carried out for HMF and VMF configurations in the paper of Strumpf [8] is given in Table 1, and these experiments and simulations are not repeated in this article. To the authors' knowledge, there appear to be no MHD simulations that have reproduced the nonlinear behavior of the rise velocity of a bubble under an applied HMF. Consequently, in this paper, we attempt to explain the mechanism based on our own MHD simulation results. The simulation cases that were undertaken correspond directly to the experiments of Wang, et al. [7]. To this purpose, the PSI-BOIL code [23,24], which was originally developed in the context of multiphase CFD simulations, was extended to include MHD effects and was applied to a single bubble rising in a conducting fluid under HMF conditions.

**Table 1.** Grid parameters for the grid dependency study of the flow in the square duct.

| *Grid Index* | *Nx* | *Ny* | *Nz* | *dx* ($\times 10^{-2}$ *a*) Equal Spacing | *dy* ($\times 10^{-2}$ *a*) | | *dz* ($\times 10^{-2}$ *a*) | |
|---|---|---|---|---|---|---|---|---|
| | | | | | Min | Max | Min | Max |
| 1.00 | 16 | 256 | 128 | 1.56 | 0.39 | 0.98 | 0.78 | 1.96 |
| 1.33 | 12 | 192 | 96 | 2.08 | 0.52 | 1.31 | 1.04 | 2.62 |
| 2.00 | 8 | 128 | 64 | 3.13 | 0.78 | 1.96 | 1.56 | 3.92 |
| 2.67 | 6 | 96 | 48 | 4.17 | 1.04 | 2.61 | 2.08 | 5.23 |
| 4.00 | 4 | 64 | 32 | 6.25 | 1.56 | 3.92 | 3.13 | 7.85 |

The outline of the paper is as follows: In Section 2, the general numerical method implemented in PSI-BOIL is outlined. The verification and validation of our MHD model are demonstrated in Section 3 and refer specifically to (i) single-phase liquid-metal flow in a channel and (ii) rising bubble simulations based on the experiments of Wang, et al. [7]. In Section 4, the MHD simulation results obtained are analyzed, and the mechanism for the nonlinear behavior of the terminal rise velocity as a function of the intensity of the applied HMF is discussed. Finally, conclusions are drawn in Section 5.

## 2. Numerical Method

### 2.1. Governing Equations

The Navier–Stokes equations for incompressible flow are defined as:

$$\nabla \cdot \boldsymbol{u} = 0, \tag{1}$$

$$\frac{\partial (\rho \boldsymbol{u})}{\partial t} + \nabla \cdot (\rho \boldsymbol{u} \boldsymbol{u}) = -\nabla p + \nabla \cdot \left\{ \mu \left( \nabla \boldsymbol{u} + (\nabla \boldsymbol{u})^T \right) \right\} + \rho \boldsymbol{g} + \boldsymbol{F}_\gamma + \boldsymbol{F}_L, \tag{2}$$

where $\rho$ is the density, $\boldsymbol{u}$ is the velocity vector, $t$ is the time, $p$ is the pressure, $\mu$ is the dynamic viscosity, $\boldsymbol{g}$ is the gravitational acceleration vector and $\boldsymbol{F}_\gamma$ and $\boldsymbol{F}_L$ are the surface tension force and the Lorentz force, respectively. In the code PSI-BOIL, the Navier–Stokes equations are discretized using a semi-implicit projection method in time [25]. The diffusion term is discretized according to the Crank–Nicolson scheme in time and the advection terms according to the Adams–Bashforth scheme. For the spatial discretization, a Cartesian finite-volume method is used in the familiar staggered variable arrangement [26], with the pressure defined at the cell center and the velocity vectors defined at the centers of the cell faces. A second-order-accurate central-difference scheme is used for the diffusion term, and a second-order scheme with a flux limiter [27] is used for the advection term.

The surface tension force is modeled according to the Continuum Surface Force (CSF) model proposed by Brackbill, et al. [28]:

$$\boldsymbol{F}_\gamma = \gamma \kappa \nabla \alpha, \tag{3}$$

where $\gamma$ is the surface tension coefficient; $\kappa$ is the curvature, which is evaluated here by use of the height function using the $3 \times 3 \times 7$ stencil proposed by López, et al. [29]; and $\alpha$ is the volume fraction of the liquid phase. The Lorentz force is defined as:

$$\boldsymbol{F}_L = \boldsymbol{j} \times \boldsymbol{B}, \tag{4}$$

where $\boldsymbol{j}$ is the electrical current density, and $\boldsymbol{B}$ is the magnetic flux density. In the present study, $\boldsymbol{B}$ is assumed to be static and equal to the applied external magnetic flux density. This is acceptable, since the induced magnetic field caused by the fluid motion is much smaller than the applied magnetic field for the applications in mind [30]. The electrical current density $\boldsymbol{j}$ is defined by Ohm's law [31]:

$$\boldsymbol{j} = \sigma(-\nabla\phi + \boldsymbol{u} \times \boldsymbol{B}), \tag{5}$$

where $\sigma$ is the electrical conductivity, and $\phi$ is the electric potential. The second term on the right hand side, $\boldsymbol{u} \times \boldsymbol{B}$, derives from the Lorentz transform [31] of Ohm's law to a moving frame of reference. The electric potential $\phi$ is calculated as follows [32]. First, the electrical current conservation law can be applied for highly conductive media:

$$\nabla{\cdot}\boldsymbol{j} = \nabla{\cdot}[\sigma(-\nabla\phi + \boldsymbol{u} \times \boldsymbol{B})] = 0, \tag{6}$$

A Poisson equation for the electric potential may then be derived by rearranging Equation (6) as follows:

$$\nabla{\cdot}(\sigma\nabla\phi) = \nabla{\cdot}(\sigma(\boldsymbol{u} \times \boldsymbol{B})). \tag{7}$$

Concerning the fluid flow, the Piecewise Linear Interface Capturing Volume Of Fluid (PLIC-VOF) method [33] is employed to track the volume fraction of the liquid phase, $\alpha$. The governing transport equation for $\alpha$ is:

$$\frac{\partial\alpha}{\partial t} + \nabla{\cdot}(\alpha\boldsymbol{u}) = 0. \tag{8}$$

The bounded conservative flux-splitting approach [34] is employed for the calculation of the advection term in order to avoid overshooting and undershooting the volume fraction, i.e., to strictly maintain the condition of $0 \leq \alpha \leq 1$. The specific implementation of the VOF method in the PSI-BOIL code is reported by Bureš, et al. [23].

In the staggered-variable arrangement, the volume fraction and the electric potential are both defined at cell centers, and the electrical current density vectors are defined at the centers of the surrounding cell faces. This is consistent with the scalar and vector representations in the method. The average density, viscosity and electrical conductivity in any control volume, which, in our case, is a cell of the underlying grid, are defined, respectively, based on the volume fraction $\alpha$ as follows:

$$\rho = \alpha\,\rho_l + (1-\alpha)\rho_g, \quad \mu = \alpha\,\mu_l + (1-\alpha)\mu_g \text{ and } \sigma = \alpha\,\sigma_l + (1-\alpha)\sigma_g, \tag{9}$$

where the subscripts $l$ and $g$ refer to the liquid and gas phases, respectively.

### 2.2. Solution Algorithm and Limitation of Time Increment

The algorithm for solving the equations is as follows.

*Step 1.* Calculate the electric potential $\phi$, Equation (7).
*Step 2.* Calculate the electrical current density $\boldsymbol{j}$, Equation (5).
*Step 3.* Calculate the Lorentz force $\boldsymbol{F}_L$, Equation (4).
*Step 4.* Calculate the surface tension force $\boldsymbol{F}_\gamma$, Equation (3).
*Step 5.* Update the velocities and the pressure by means of the projection method, Equations (1) and (2).
*Step 6.* Update the volume fraction $\alpha$, Equation (8).
*Step 7.* Advance the time step, and go back to *Step 1*.

The solution of the Poisson equation for the electric potential (*Step 2*) is obtained using the bi-conjugate gradient stabilized (BiCGSTAB) method [35], whereas the Poisson equation for the pressure (*Step 5*) is solved using the conjugate gradient (CG) method [36]. BiCGSTAB is required because the electrical conductivities for the liquid metal and the gas differ by more than twenty orders of magnitude. The algebraic multigrid method [37] is applied to the Poisson equations for the electric potential and the pressure to accelerate their convergence.

The time increment $\Delta t$, which is limited both by the CFL condition and the stability condition for the surface tension, is given by:

$$\Delta t = \min\left( c_{CFL} \frac{\Delta x}{|\boldsymbol{u}|_{\max}}, \left( \frac{\rho_g \Delta x^3}{2\pi\gamma} \right)^{\frac{1}{2}} \right),$$ (10)

where $c_{CFL}$ is a dimensionless constant representing a safety factor, and $\Delta x(= \Delta y = \Delta z)$ is the grid spacing. In this paper, we chose $c_{CFL} = 0.25$. The entire procedure is an MHD extension of the algorithm implemented in PSI-BOIL to calculate the fluid flow field [23].

*2.3. Assumptions*

For clarity, the assumptions used in the numerical methods employed are summarized here. The working fluid is modeled as incompressible, and the presence of any surfactant is neglected based on the assumption that the working fluid is not contaminated and that the surface tension coefficient is therefore constant. The magnetic field $\boldsymbol{B}$ is assumed to be uniform and static everywhere at the value of the applied external field. This assumption is valid, since the magnetic Reynolds number $Rm << 1$, meaning that internal distortions in the magnetic field strength are negligible. The electrical current conservation law, $\nabla \cdot \boldsymbol{j} = 0$, is also adopted, since liquid metals are generally very good electrically conductive media.

## 3. Verification and Validation

The numerical method described in the previous sections was implemented into the PSI-BOIL code [23,24], which was originally developed for two-phase flow simulations, including phase change phenomena. In this section, two cases of verification and validation are presented: the first refers to single-phase flow, and the second refers to two-phase flow, both of which are subject to an externally applied magnetic field.

*3.1. Single-Phase Liquid-Metal Flow in a Square Channel*

Single-phase liquid-metal flow in a square channel under the influence of a static magnetic field was computed. The results were compared with the analytical solution proposed by Shercliff [38], and the experimental data were measured by Hartmann and Lazarus [39]. In the experiment, a rectangular channel was filled with mercury, and the flow was driven by an applied pressure drop. A uniform and static magnetic field was applied in a direction lateral to the axial direction of the channel, as seen in Figure 1a. The mass flow rate was measured for different intensities in the magnetic field. We focus here on the experiment *K 33*, which Shercliff himself used as a validation case, with a square channel with electrically insulating walls and with dimensions of the sectional area being $1.14 \times 1.16$ mm$^2$ and the length being 140 mm. The Hartmann number, which represents the ratio of the magnetic force to the viscous force, for this problem is defined as follows:

$$Ha_{channel} = aB_0 \sqrt{\frac{\sigma_l}{\mu_l}},$$ (11)

where $a$ is the half-width of the channel, and $B_0$ is the intensity of the imposed magnetic field. In the *K 33* experiment, measurements were undertaken over the range of $0 \le Ha_{channel} \le 18$. Note that we performed simulations over the wider range of $0 \le Ha_{channel} \le 50$ in order to compare them with not only the experiment but also with the analytical solution [38].

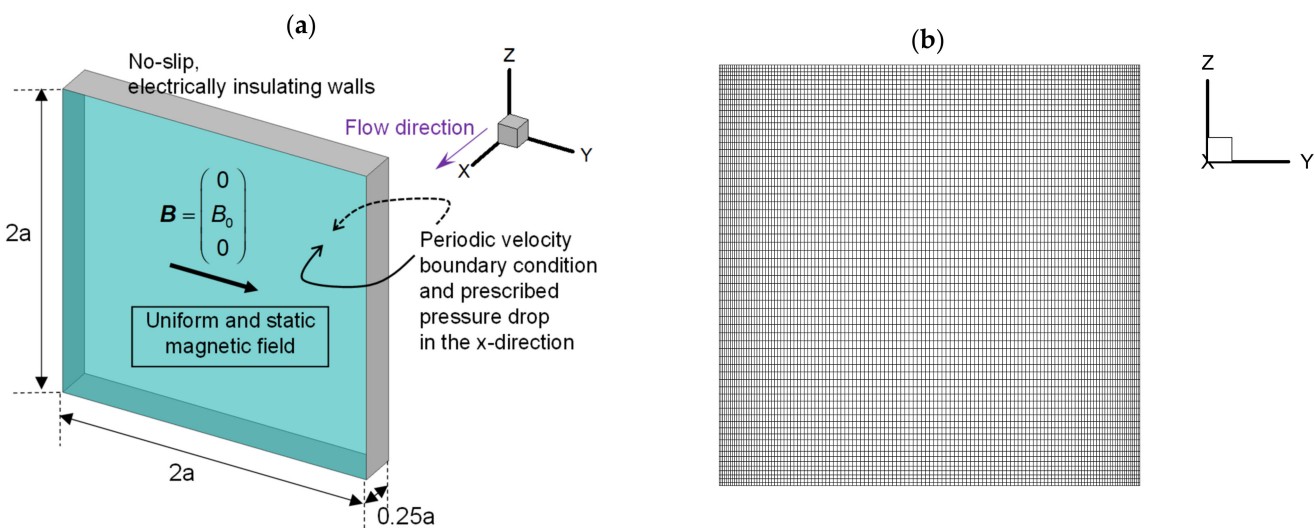

**Figure 1.** Computational domain and boundary conditions for single-phase liquid-metal flow in a square channel (**a**), and the computational grid for *Grid index* = 2.00 in the Y–Z plane (**b**).

The size of the computational domain, the boundary conditions, and the orientation of the coordinate system are all illustrated in Figure 1a: the channel width and height were set to 2*a*, and the axial domain length was set to 0.25 *a*. No-slip velocity boundary conditions were applied at the walls surrounding the channel, and these were modeled as electrically insulating materials. Periodic velocity boundary conditions were employed at the boundaries in the *x*-direction, where a prescribed pressure drop was imposed. A uniform and static magnetic field of intensity $B_0$ was applied in the lateral direction, which is also indicated in Figure 1. The number of cells was chosen as 8 × 128 × 64 in the *x*-, *y*- and *z*-directions, respectively. The cell size in the *x*-direction was constant at $3.13 \times 10^{-2}$ *a*, whereas stretched cell sizes were used in the *y*- and *z*-directions, as depicted in Figure 1b, in order to explicitly resolve the wall boundary layers. The cell sizes $dy$ and $dz$, in the other directions, were in the ranges of $7.81 \times 10^{-3}$ $a \le dy \le 1.96 \times 10^{-3}$ $a$ and $1.56 \times 10^{-2}$ $a \le dz \le 3.94 \times 10^{-2}$ $a$, respectively.

### 3.1.1. Grid Dependency Study

A grid dependency study was carried out based on the grid convergence index method [40] to serve as a verification test of the model's implementation. Five cases with different cell sizes were computed for $Ha_{channel}$ = 50. The cell size in the *x*-direction remained constant, whereas stretched cells were employed in the *y*- and *z*-directions within the ranges stated in the previous paragraph. The grid parameters are listed in Table 1, where *Grid index* represents the notional non-dimensional grid size; *Nx*, *Ny* and *Nz* are the number of cells in the *x*-, *y*- and *z*-directions, respectively; and *dx*, *dy* and *dz* are the cell sizes in each direction in units of *a*. For illustrative purposes, the grid corresponding to *Grid index* = 2.00 is shown in Figure 1b.

The computations were continued until steady-state conditions were attained. The computed mean velocity $v_0$ for each case is shown in Figure 2 as a function of the *Grid index*. The vertical axis is $ka^2/v_0$, where $k\left(= \frac{dp}{dx}\right)$ is the pressure drop applied to the channel. The analytical solution [38] is also depicted in Figure 2 in comparison with the computed result. The exponent of the fitted curve, which indicates the accuracy of the numerical scheme in space, is 1.9. Since all the governing equations are discretized using second-order schemes in space, the 1.9th-order dependency is reasonable.

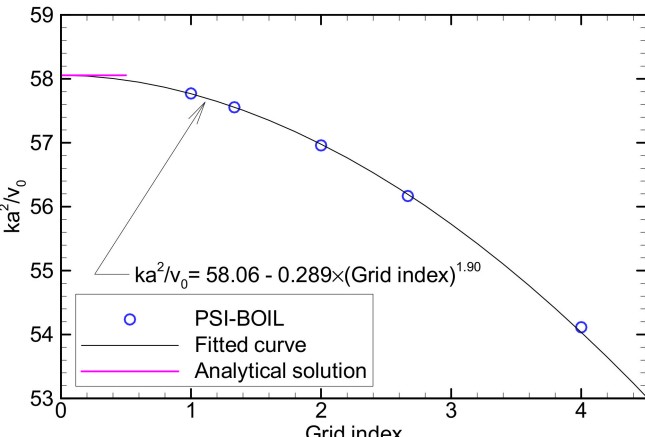

**Figure 2.** Grid dependency study for $Ha_{channel}$ = 50, indicating the 1.9th-order accuracy in spatial discretization.

3.1.2. Comparisons against Analytical Solutions and Experiments

Eight simulation cases for different $Ha_{channel}$ were simulated, namely $Ha_{channel}$ = 0, 5, 10, 15, 20, 30, 40 and 50. The computational grid with *Grid index* = 1.00 was used for all the simulations, since, according to Figure 3, this is clearly in accord with the asymptotic limit. Once the solution converged, all the variables (except the pressure) were constant in the $x$-direction. Thus, we only present the distribution of the variables in a single $x$-constant plane, all of which were non-dimensionalized as follows: $x' = \frac{x}{a}$, $u' = u\sqrt{\frac{\rho}{ak}}$, $F' = \frac{F}{a^3 k}$, $\phi' = \phi \frac{\sqrt{\rho}}{B_0\sqrt{a^3 k}}$ and $j' = j \frac{\sqrt{\rho}}{\sigma B_0 \sqrt{ak}}$, where the variable with the superscript $'$ indicates a non-dimensional value.

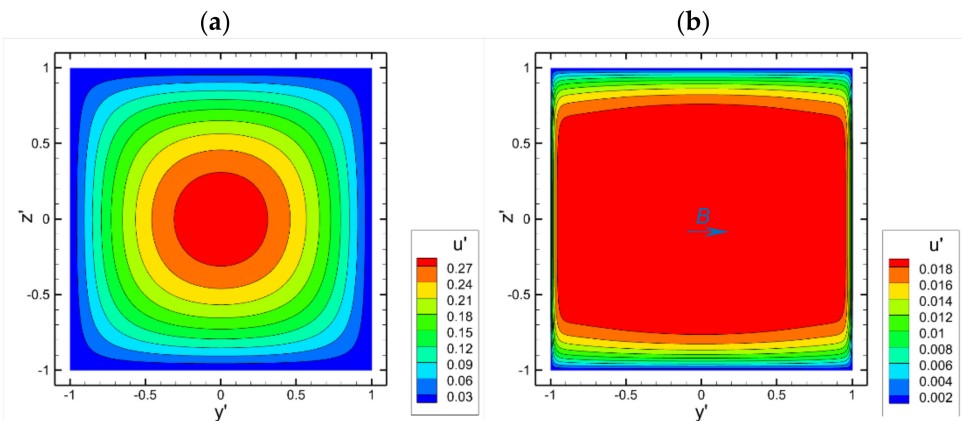

**Figure 3.** Distribution of the non-dimensional axial velocity $u'$ in a constant $x'$ plane for $Ha_{channel}$ = 0 (**a**) and $Ha_{channel}$ = 50 (**b**).

The distribution of the axial velocity for $Ha_{channel}$ = 0 and 50, representing the extreme cases, are compared in Figure 3. The lengths and velocities are non-dimensionalized according to the base parameters, i.e., $y' = \frac{y}{a}$, $z' = \frac{z}{a}$, $u' = u\sqrt{\frac{\rho}{ak}}$. The result for $Ha_{channel}$ = 0 is a typical profile for laminar flow in a square channel, whereas a steep velocity gradient is observed near the walls at $y$-min and $y$-max for $Ha_{channel}$ = 50. The velocity components in the $y$ and $z$ directions are exactly zero in both cases. Figure 4 shows the normalized distributions of (a) the Lorentz force, (b) the electric potential and (c) the magnitude of the electric current density and current path. For clarity, the electric current paths are depicted only in the left half of the domain. The Lorentz force, which acts in the $x$-direction, is

strongest near the *y*-min and *y*-max walls, induced by the high value of the electric current density there.

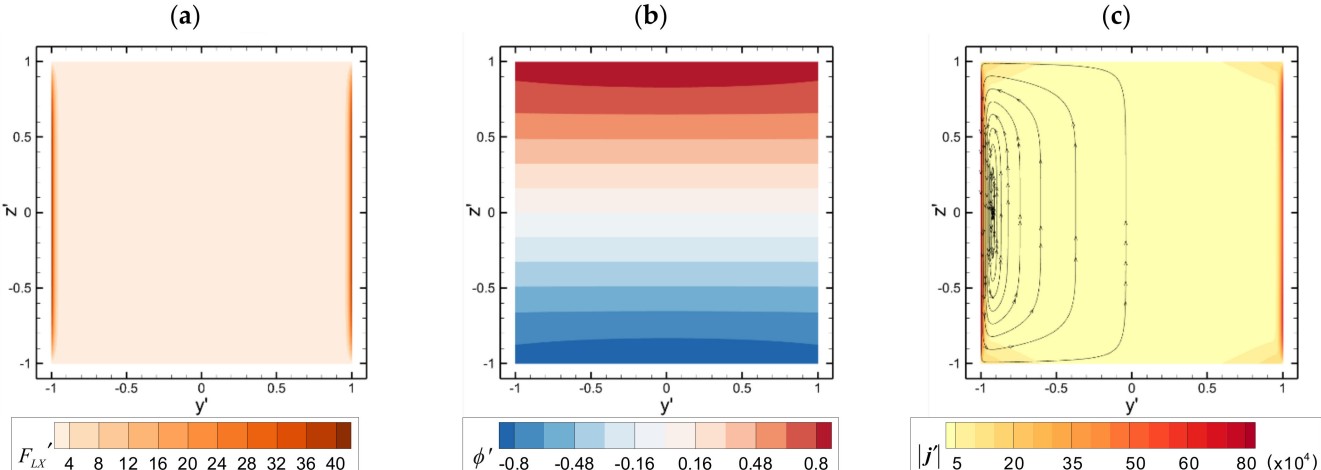

**Figure 4.** Normalized distributions of (**a**) the Lorentz force in the *x*-direction $F_{LX}{}'$, (**b**) the electric potential $\phi'$ and (**c**) the magnitude of the electric current density $\left|j'\right|$, and the electric current vector $j'$ (lines drawn only in the left half of the domain for clarity).

In Figure 5, the computed mean velocity $v_0$ is compared with the analytical solution [38] as well as with the experimental measurements [39] for different Hartmann numbers. Our simulation results are in very good agreement with both the analytical solution and the measurements.

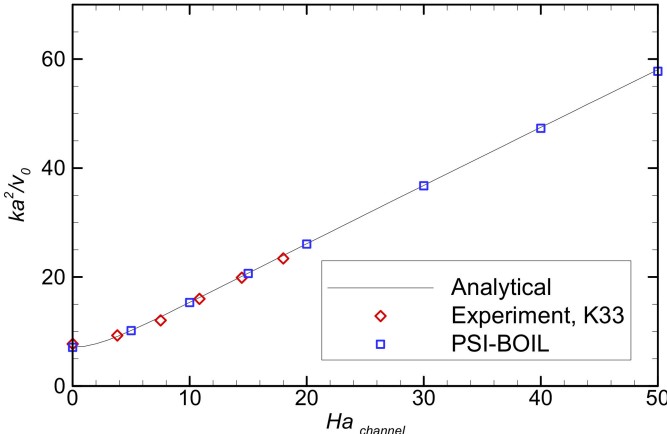

**Figure 5.** Comparisons of the inverse of the non-dimensionalized mean velocity $ka^2/v_0$ between the analytical solution [38], the experiment [39] and our CFD simulations obtained using PSI-BOIL.

### 3.2. Rising Single Bubble in Liquid Metal

A series of simulations of a single bubble rising in liquid metal was performed for the verification and validation tests of our MHD model introduced into PSI-BOIL. The conditions of the simulation were taken from the experiments performed by Wang, et al. [7]. In these experiments, single argon bubbles were injected through a nozzle into a rectangular tank filled with the liquid metal GaInSn. The tank was constructed of acrylic glass with dimensions $60 \times 60 \times 200$ mm$^3$ in both lateral directions and height, respectively. The tank was subject to a homogeneous, static and transverse magnetic field, and the rise velocity of the bubbles was measured using a UDV technique [20]. Due to the opacity of the liquid metal, the bubble shape and bubble rise path could not be measured directly, though the bubble volume was estimated based on the volume flow rate of the argon ejected from the

nozzle and the number of bubbles counted during the measurement period. The bubble diameter ranged from 3.1 mm to 5.6 mm, which corresponds to an Eötvös number of $Eo$ = 1.12 to 3.67. The maximum applied magnetic field strength was 2 $T$. The material properties for argon and GaInSn were taken from the paper of Wang, et al. [7] and are listed in Table 2.

**Table 2.** Physical properties of materials for validation test.

| Material | Density (kg/m$^3$) | Viscosity (Pa.s) | Electrical Conductivity (1/$\Omega$m) | Surface Tension Coefficient (N/m) |
|---|---|---|---|---|
| Ar | 1.654 | $1.176 \times 10^{-5}$ | $1.000 \times 10^{-15}$ | 0.553 |
| GaInSn | $6.362 \times 10^3$ | $2.200 \times 10^{-3}$ | $3.270 \times 10^6$ | |

Figure 6 shows the orientation of the coordinate system, the boundaries of the computational domain, the initial bubble size and the direction of the applied magnetic field. Both the $x$- and $y$-axes are horizontal, and the $z$-axis points upward with the origin at the center of the bottom plane. A uniform and static magnetic field, $B_0$, was applied in the positive $x$-direction. The domain width was $6d \times 6d$, where $d$ is the diameter of the initial spherical bubble. The height of the domain, $LZ$, was set in such a way that a pseudo-steady-state could be obtained for the bubble rise velocity in all cases. In practical terms, this translated as $LZ = 24d$ for the simulation cases with $B_0 > 0$ and $LZ = 48d$ or $96d$ for the cases in which $B_0 = 0$. The domain width $6d \times 6d$ is the same as the simulation setup of Jin et al. [17]. In this paper, we simulated the bubbles in the range of $1.12 \leq Eo \leq 3.67$, and the domain width $6d \times 6d$ corresponds to $19 \times 19$ mm$^2$ for $Eo$ = 1.12 and $34 \times 34$ mm$^2$ for $Eo$ = 3.67, which is smaller than the experimental setup of $60 \times 60$ mm$^2$ [7] but which has a similar order of dimension. The walls surrounding the computational domain were assumed to be non-slip for the velocity field ($\boldsymbol{u} = 0$, $\partial p / \partial n_w = 0$, where $\boldsymbol{n}_w$ is the wall normal vector), and they were also assumed to be electrically insulating ($\boldsymbol{j} \cdot \boldsymbol{n}_w = 0$). A spherical bubble was initially placed with its center at the coordinate (0, 0, 2$d$) above the center of the bottom face in the stagnant liquid GaInSn. The computational grid consisted of uniform cubes, and a fixed grid was used, i.e., neither a moving grid nor a grid refinement technique was employed.

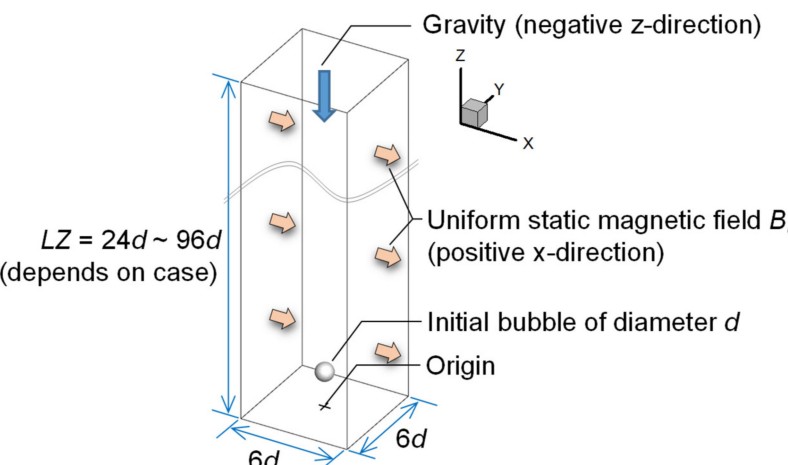

**Figure 6.** Computational domain size, the direction of the applied magnetic field and the orientation of the coordinate system for the rising bubble simulation. The domain is surrounded by no-slip walls.

3.2.1. Grid Dependency Study

First, a grid dependency study was performed for the selected flow cases, as listed in Table 3. Case G-B0 refers to a zero magnetic field, whereas G-B2 is the case for which $B_0$ = 1.97 T, which is the maximum intensity of the magnetic field featured in the experiment

of Wang, et al. [7]. The bubble diameter, based on the bubble volume, was 5.6 mm for both cases. The Morton number, *Mo*, and the Hartmann number, *Ha*, used in the table are defined as follows:

$$Mo = \frac{g\mu_l^4(\rho_l - \rho_g)}{\rho_l^2\gamma^3}, \quad Ha = dB\sqrt{\frac{\sigma_l}{\mu_l}}, \tag{12}$$

**Table 3.** Simulation cases for the grid dependency study.

| Case | $B_0$ (T) | $d$ (mm) | Eo | Mo | Ha | LZ |
|------|-----------|----------|-----|-----|-----|-----|
| G-B0 | 0 | 5.6 | 3.67 | $2.38 \times 10^{-13}$ | 0 | 48$d$ |
| G-B2 | 1.97 | 5.6 | 3.67 | $2.38 \times 10^{-13}$ | 425 | 24$d$ |

An extended computational domain with *LZ* = 48*d* was needed for the simulation case G-B0 in order to assure pseudo-steady-state conditions at the end of the transient. Moreover, *LZ* = 24*d* proved to be high enough for G-B2, since steady-state conditions were observed to be achieved over a short distance due to the slower rising bubble, the details of which are presented later. Five different grid spacings were adopted for this study, as listed in Table 4. In each case, the computational grid consisted of uniform cubes of side $\Delta x$.

**Table 4.** Grid parameters for the grid dependency study of the rising bubble simulation.

| *Grid Index* | *d*/$\Delta x$ |
|--------------|----------------|
| 1.00 | 21.3 |
| 1.33 | 16.0 |
| 1.60 | 13.3 |
| 2.00 | 10.7 |
| 2.67 | 8.0 |

The computed bubble rise velocity as a function of time for the case GB-0 is shown in Figure 7a. The rise velocity at time step *n* was calculated according to $(Z_{t^{n+1}} - Z_{t^{n-1}})/(t^{n+1} - t^{n-1})$, where $Z_t$ is the *z*-coordinate of the bubble centroid at time *t*, defined as:

$$Z_t = \frac{\sum\limits_{i=1}^{ncells} \alpha_i V_i z_i}{\sum\limits_{i=1}^{ncells} \alpha_i V_i}, \tag{13}$$

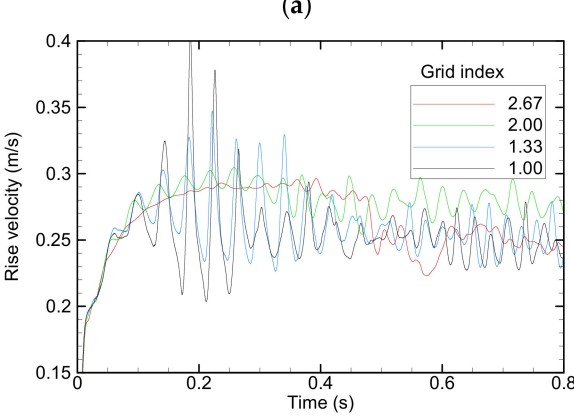
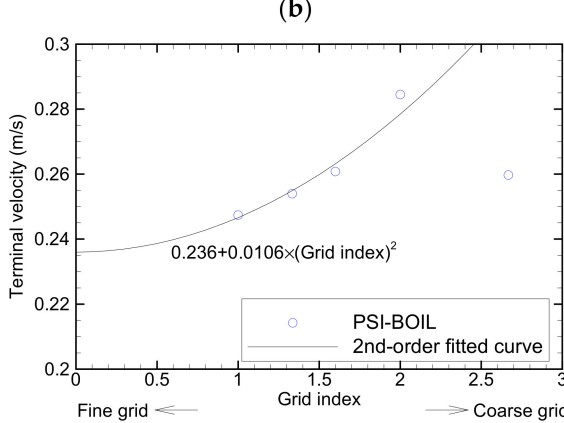

**Figure 7.** Grid dependency study for Case G-B0: evolution of the rise velocity (**a**), and the grid dependency of the terminal velocity (**b**).

Here, $V_i$ and $z_i$ are the volume and $z$-coordinate of the computational cell center indexed $i$, respectively, and *ncells* is the total number of cells. The rise velocities did not attain steady-state conditions for all the grids adopted, since the bubble shape evolved with time, and the trajectories were not strictly vertical, as shown in Figure 8. The time-averaged rise velocity was calculated as follows:

$$u_T = \frac{Z_{t_2} - Z_{t_1}}{t_2 - t_1},\qquad(14)$$

where $Z_{t_1}$ and $Z_{t_2}$ are the $z$-coordinates of the bubble centroid at times $t_1$ and $t_2$, respectively. For the case of GB-0, $t_1$ and $t_2$ were set to 0.6 s and 0.8 s, respectively, and the grid dependency of the time-averaged velocity is displayed in Figure 7b. The terminal velocity for the coarsest grid was considered to be out of the asymptotic region, and the second-order fitted curve was drawn neglecting this "rogue" result. Note that all the data points are not exactly on the fitted curve because the simulations presented here were unsteady, which made the estimation of the discretization error more challenging than those of the steady flow simulations due to time discretization errors, as mentioned in the review paper of Eça, et al. [41]. The asymptotic behavior indicates that the order of accuracy in space is indeed close to being second-order, on the basis of the fitted curve.

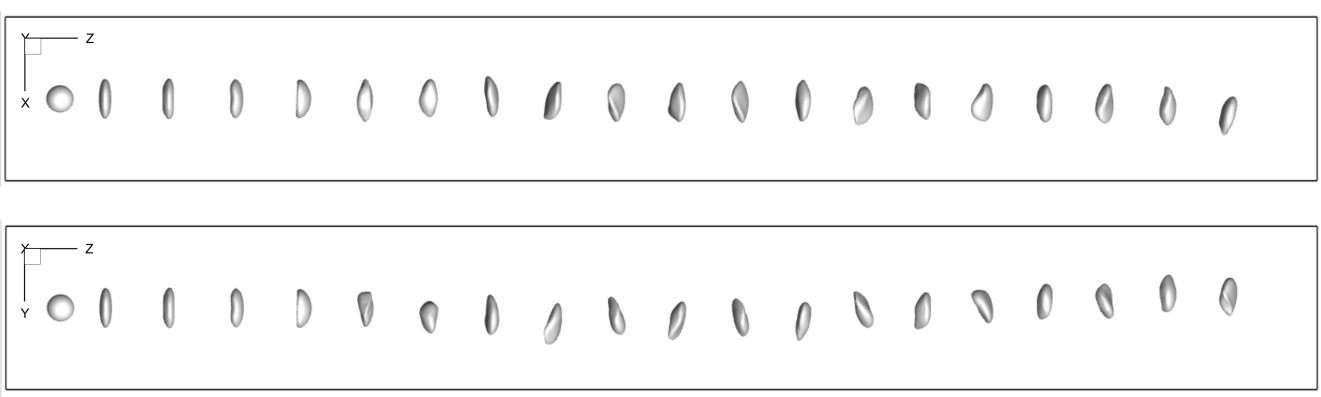

**Figure 8.** Evolution of bubble shape (drawn horizontally for convenience) for G-B0 with *Grid index* = 1.00. The bubble shapes are drawn at time intervals of 0.05 s.

The time–history of the rise velocity for case G-B2 is shown in Figure 9a. As can be noted, the rise velocities attained steady-state values after $t = 0.8$ *s*, except for the case of the coarsest grid, which, again, was considered to be out of the asymptotic region, which was the same for simulation G-B0 with this same *Grid index*. The terminal rise velocity, which is calculated here from Equation (14) for $t_1 = 0.8$ s and $t_2 = 1.0$ s, is drawn in Figure 9b. In this figure, near-second-order behavior is indicated from the fitted curve, with the value from the coarsest grid case again being neglected due to it being out of the asymptotic region. The bubble shapes for different grids are compared in Figure 10. In general, the shapes resemble each other, especially for the finest grids, with *Grid index* = 1.00 and 1.33. Based on these results, a grid spacing of $\Delta x = d/16$ (*Grid index* = 1.33) was adopted for all the simulations described hereafter in order to obtain trustworthy results and to economize CPU time.

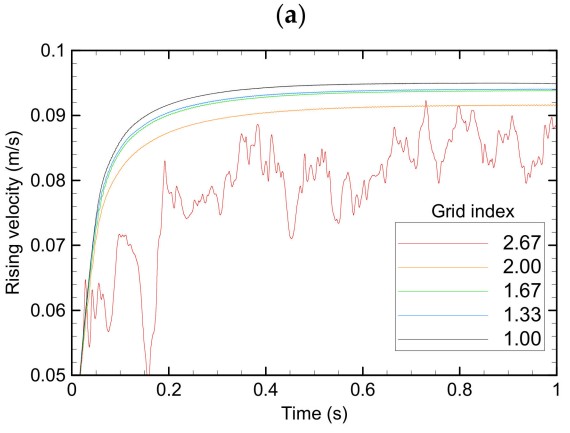

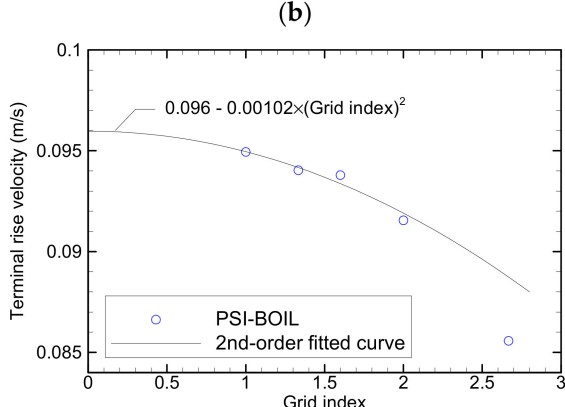

**Figure 9.** Grid dependency study for Case G-B2: (**a**) time evolution of bubble rise velocity, (**b**) terminal rise velocity as a function of *Grid index*. The terminal rise velocity converges at 2nd-order accuracy.

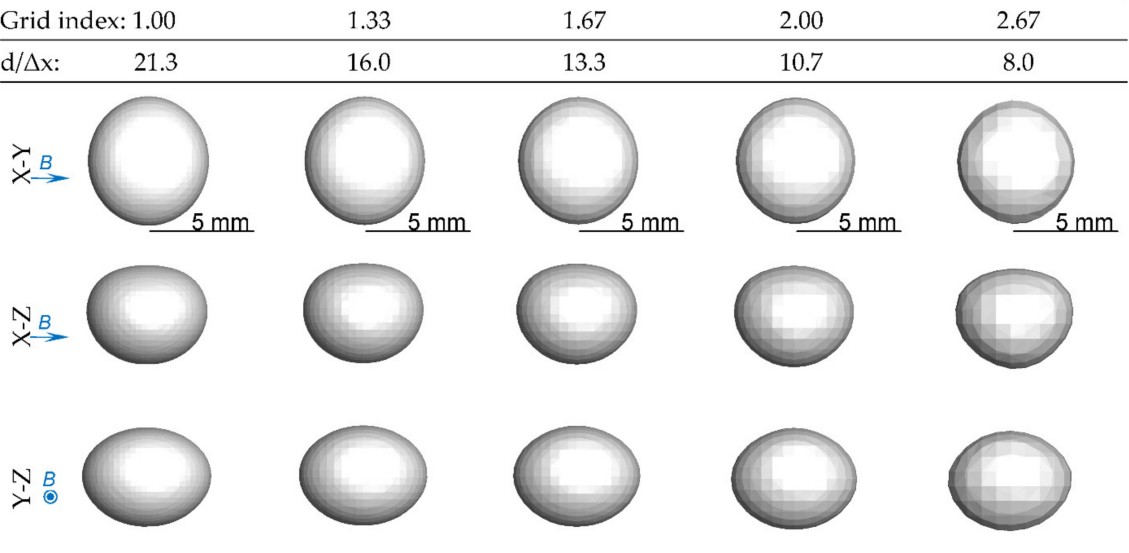

**Figure 10.** Grid dependence of the bubble shape in the three coordinate planes at $t = 1$ s for case G-B2.

### 3.2.2. Comparison with the Measurements

In order to validate the installed MHD model in PSI-BOIL, 20 simulation cases were undertaken. These include five cases involving different initial bubble diameters and six cases of different magnetic field intensities, as listed in Table 5. Note that the computational domain height *LZ* (see Figure 6) depends on the specific simulation case. The grid spacing was set at $\Delta x = d/16$ for all the cases on the basis of the grid dependency study described above.

**Table 5.** List of validation cases: $Mo = 2.38 \times 10^{-13}$ for all cases.

| *d* (mm) | *Eo* | **Magnetic Field $B_0$ and Domain Height *LZ*: ($B_0$(T), *LZ* (*d*))** |
|---|---|---|
| 3.10 | 1.12 | (0, 96), (0.14, 48), (0.28, 48), (0.56, 24), (1.12, 24), (1.97, 24) |
| 3.40 | 1.35 | (0, 96) |
| 4.57 | 2.44 | (0, 48), (0.14, 48), (0.28, 48), (0.56, 24), (1.12, 24), (1.97, 24) |
| 5.15 | 3.10 | (0, 48) |
| 5.60 | 3.67 | (0, 48), (0.14, 48), (0.28, 24), (0.56, 24), (1.12, 24), (1.97, 24) |

The computed terminal rise velocities for the cases with $B_0 = 0$ are compared with the measurements of Wang, et al. [7] and the Tomiyama correlation [42] in Figure 11. The bubble diameters were all in the inviscid flow regime according to the Tomiyama correlation, for which the terminal velocity $u_T$ is given by:

$$u_T = \sqrt{\frac{2\gamma}{\rho_l d} + \frac{(\rho_l - \rho_g)gd}{2\rho_l}}.$$

(15)

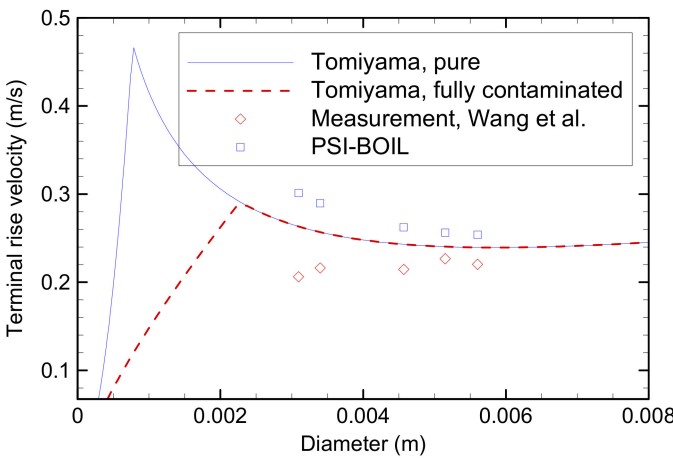

**Figure 11.** Comparisons of terminal rise velocity for $B_0 = 0$ between the Tomiyama correlations [42], the measurements of Wang, et al. [7] and the simulated values obtained in this study.

The correlation was originally proposed by Marrucci, et al. [43] and, in a slightly different form, in the model of Mendelson [44]. However, these are not theoretically derived from the Navier–Stokes or Euler equations, and their validity is still open to question, according to Tomiyama, et al. [42]. Our computed terminal velocities in Figure 11 are higher than those obtained from the Tomiyama correlation in all the cases considered, but the trend is the same, i.e., a decrease in the terminal velocity with an increase in bubble diameter. In contrast, the measured velocities of Wang, et al. [7] are consistently lower than those derived from the Tomiyama correlation in all cases, and even the trend is opposite, i.e., the measured data show a slight increase in the terminal velocity with an increase in bubble diameter, whereas the Tomiyama correlation predicts a slight decrease. The different trends may indicate that the liquid metal in the experiment might have been contaminated even though special attention was paid in regard to this issue [7], although such a conclusion must be regarded as speculative at this stage. In other words, the measured data display a similar trend to that predicted from the Tomiyama correlation for a fully contaminated fluid rather than the trend for a pure fluid. Haas, et al. [45] also speculated that the liquid metal used in the experiment of Wang, et al. [7] was contaminated in their review paper [45] on the subject. Wang, et al. [7] discussed in their paper that the lower rise velocity measured in their experiment compared with that of the Tomiyama correlation was caused by the lower surface tension coefficient under the experimental conditions compared with that measured under static conditions. However, we consider that Wang et al.'s hypothesis cannot explain the opposite trend of the velocity, since the lower surface tension coefficient only shifts the Tomiyama correlation lower but does not change the trend. Thus, we consider that the liquid metal used in Wang et al.'s experiment was indeed contaminated.

The mechanism for the decrease in rise velocity due to contamination is considered to be caused by the Marangoni effect, as proposed by Frumkin and Levich [46], i.e., impurities accumulate at the bubble interface, and they are transported from the bubble front/top to the rear/bottom along the interface due to the main flow. As a result, the surface tension coefficient on the front side becomes stronger than that on the rear side because impurities decrease the surface tension coefficient in general. Since a liquid with a high surface tension

coefficient pulls the surrounding liquid more strongly than one with a low surface tension, a force pointing from the rear to the front along the interface of the bubble appears. The force points in the opposite direction of the main flow, which results in an increase in the drag of the rising bubble.

Comparisons of measured and simulated terminal velocities for different magnetic field strengths and initial bubble diameters are given in Figure 12. For $d$ = 3.10 mm (Figure 12a), large discrepancies can be seen between the measurements and calculations for cases with lower $B_0$ values, and we have already alluded to the discussion accompanying Figure 11 for $B_0$ = 0. The influence of the magnetic field on the terminal velocity for $d$ = 3.10 mm shows a similar tendency, i.e., the velocity increases or does not change with an increase in $B_0$ in the lower magnetic intensity range ($B_0 \leq 0.14$ T), but it decreases with increasing $B_0$ in the upper range ($B_0 > 0.28$ T). The measurement and simulation agree well in the range of 1.12 T $\leq B_0 \leq$ 1.97 T. This is also true for the cases in which $d$ = 4.57 mm and 5.60 mm. Better agreement for higher values of $B_0$ may imply that the rise velocity is influenced more by the Lorentz force than by any fluid contamination.

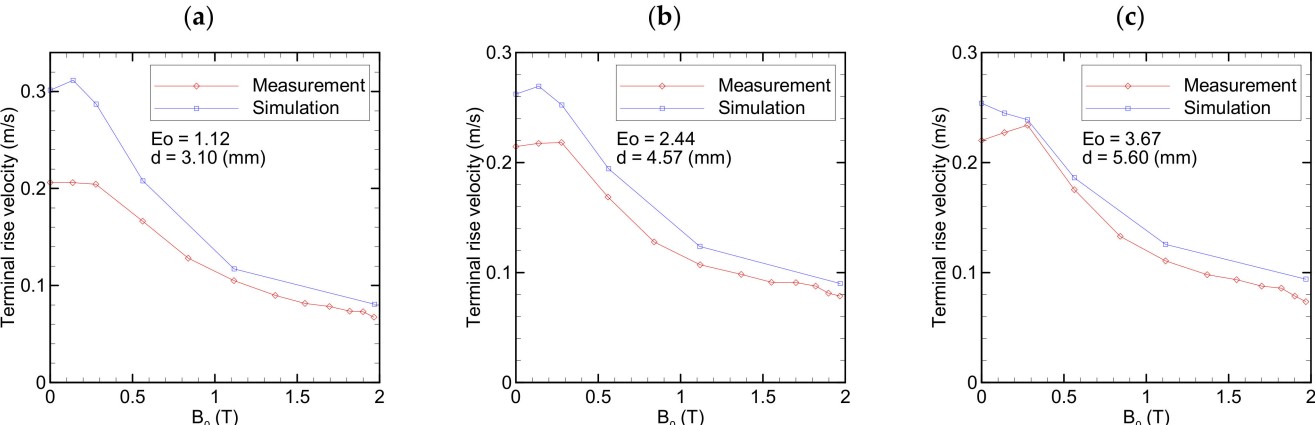

**Figure 12.** Comparison of terminal rise velocities as functions of $B_0$ for $d$ = 3.10 mm (**a**), 4.57 mm (**b**) and 5.60 mm (**c**).

The result of the simulation trend with a lower magnetic field intensity ($B_0 \leq 0.28$ T) in Figure 12 is different from that of the measurement. The nonlinear behavior of the rise velocity is a consequence of two factors: (i) a decrease in drag due to the Lorentz force pointing in the opposite direction to the flow field in the Y–Z plane, which prevents vortex shedding, and (ii) an increase in drag due to the Lorentz force pointing upward in the X–Z plane, which induces a larger wake. These two factors, which are later explained in Section 4.4.1 using Figures 21 and 22, almost balance each other for the lower magnetic intensity, and the discrepancy between the measurement and the simulation is magnified since these forces are small. Nonetheless, the feature of a flat velocity profile in the region of lower magnetic field intensity ($B_0 \leq 0.28$ T) is reproduced by the simulation.

Since the experiment of Wang, et al. [7] was selected for the simulation cases in this study, the range of $Eo$ was limited to $1 < Eo < 4$, and the influence of HMF on a rising bubble for $Eo < 1$ or $Eo > 4$ was not evaluated. According to the measurement of Mori [1], a bubble with higher $Eo$ (=10) showed a monotonic decrease in rise velocity with an increase in HMF, which is the same tendency as our simulation result for $Eo$ = 3.67, as depicted in Figure 12c. Moreover, both the measurement and simulation of a rising bubble with $Eo << 1$ are considered to be not straightforward because of its small size, which can be a future research topic.

## 4. Results and Discussion

In this section, we analyze the simulation results for $Eo$ = 2.44, which is considered to be a typical case, and we discuss: (i) the influence of the magnetic field on the flow and

(ii) the nonlinear behavior of the rise velocity as a function of the magnetic field strength. The simulation conditions were the same as those for the validation case, as seen in the third row of Table 5.

### 4.1. Steady or Unsteady Bubble Rise Velocity and Bubble Shape

The computed rise velocities for $Eo$ = 2.44 are shown in Figure 13a as functions of time. A constant rise velocity was attained for the cases in which $B_0 \geq 0.56$ T, and a periodic condition prevailed for $B_0 = 0.28$ T. In the cases of $B_0 = 0$ and $B_0 = 0.14$ T, the rise velocity displayed periodic behavior in the (approximate) range of 0.1 s $\leq t \leq$ 0.4 s, but it changed to a higher frequency fluctuation mode at later times. The time-averaged velocity is shown in Figure 13b, which was calculated from Equation (14) with $(t_1, t_2)$ = (0.5 s, 0.7 s) for cases in which $B_0 \leq 0.28$ T and (0.35 s, 0.45 s) for cases in which $B_0 \geq 0.56$ T. As argued in Section 3, the discrepancy between the measurement and the calculation is considered to be caused by the contamination/oxidation of the liquid metal in the experiment. Figure 13c shows the bubble Reynolds number as a function of $N/C_d$. The peak Reynolds number appeared at $N/C_d = 0.2$ in the simulation and at $N/C_d = 0.7$ in the experiment of Wang, et al. [7]. In the experiments of Strumpf [8], the peak Reynolds number was observed to be around 1.0, but the tank used in the experiment was narrow, with dimensions of $144 \times 12 \times 200$ mm$^3$ ($L$, $W$, $H$). This might have had an influence on the terminal rise velocity due to the proximity of the bubble to the side walls and the influence of the boundary layers at such a low Reynolds number.

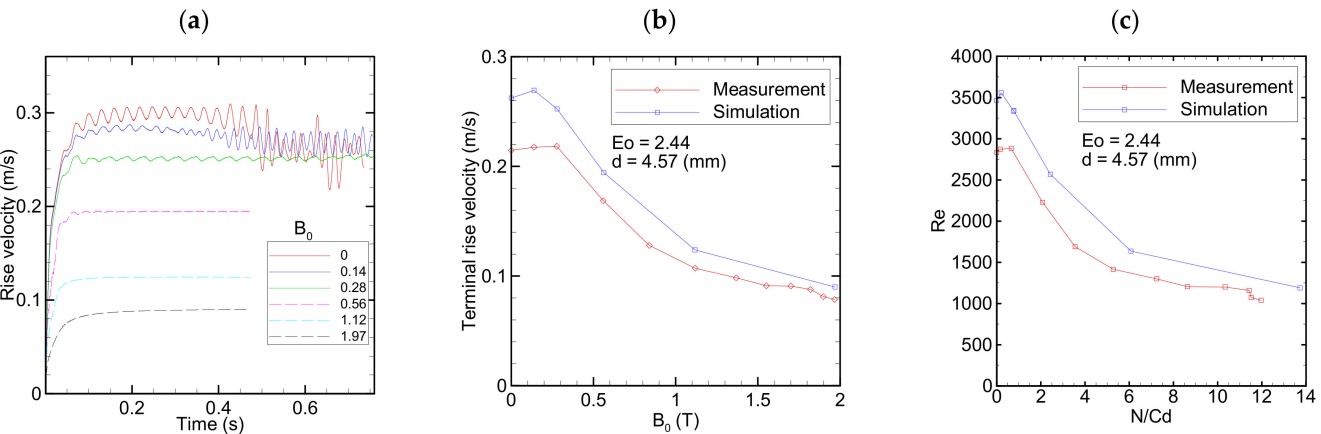

**Figure 13.** Computed rise velocities for different magnetic field strengths for $Eo$ = 2.44: (**a**) the time history of the rise velocity, (**b**) the time-averaged rise velocity as a function of the magnetic field intensity and (**c**) the bubble Reynolds number as a function of $N/C_d$.

The computed bubble shapes are shown in Figure 14. Those for $B_0 = 0$ T, 0.14 T and 0.28 T change with time, and the shape displayed in the figure is a representative snapshot at $t = 0.6$ s. The bubble shapes for $B_0 \geq 0.56$ T are all steady in time; thus, the influence of the magnetic field on the bubble shape can be evaluated from these snapshots. In the X–Y plane (top view), the bubble for $B_0 = 0.28$ T is slightly elongated in the $y$-direction, as recognized by the comparison with the circle drawn with a dashed red line. The shapes for $B_0 = 0.28$ T in the X–Z and Y–Z planes display an opposite characteristic; the bottom half is rounder than the top half in the X–Z plane, but this feature is reversed in the Y–Z plane. As the magnetic field increases from 0.28 T to 1.97 T, the bubble approaches a more spherical shape.

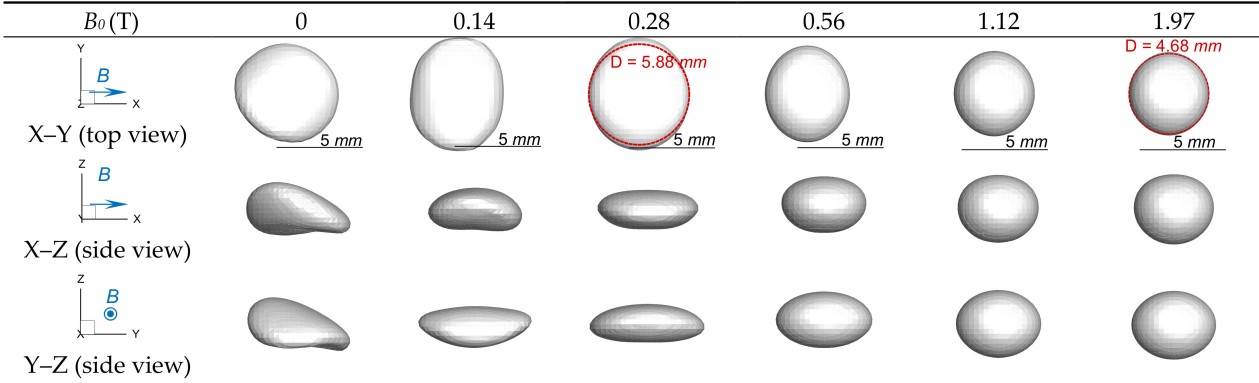

**Figure 14.** Calculated bubble shapes for different applied magnetic fields for $Eo = 2.44$. The snapshots were taken at $t = 0.6$ s for $B_0 = 0$ T, 0.14 T and 0.28 T, and at $t = 0.4$ s for $B_0 = 0.56$ T, 1.12 T and 1.97 T. The bubble shapes for $B_0 \geq 0.56$ T are all steady in time, and the bubble approaches a more spherical shape as the magnetic field increases from 0.28 T to 1.97 T.

The evolving unsteady bubble shapes for $B_0 = 0$ T, 0.14 T and 0.28 T are shown in Figure 15. Those for $B_0 = 0$ are asymmetric, and the trajectory is not strictly vertical. In contrast, those for $B_0 = 0.14$ T and 0.28 T change but remain laterally symmetric in the X–Z and Y–Z planes, and the trajectory is vertical. The symmetric shape is considered to be a result of the applied magnetic field, and further details are discussed in the following sections.

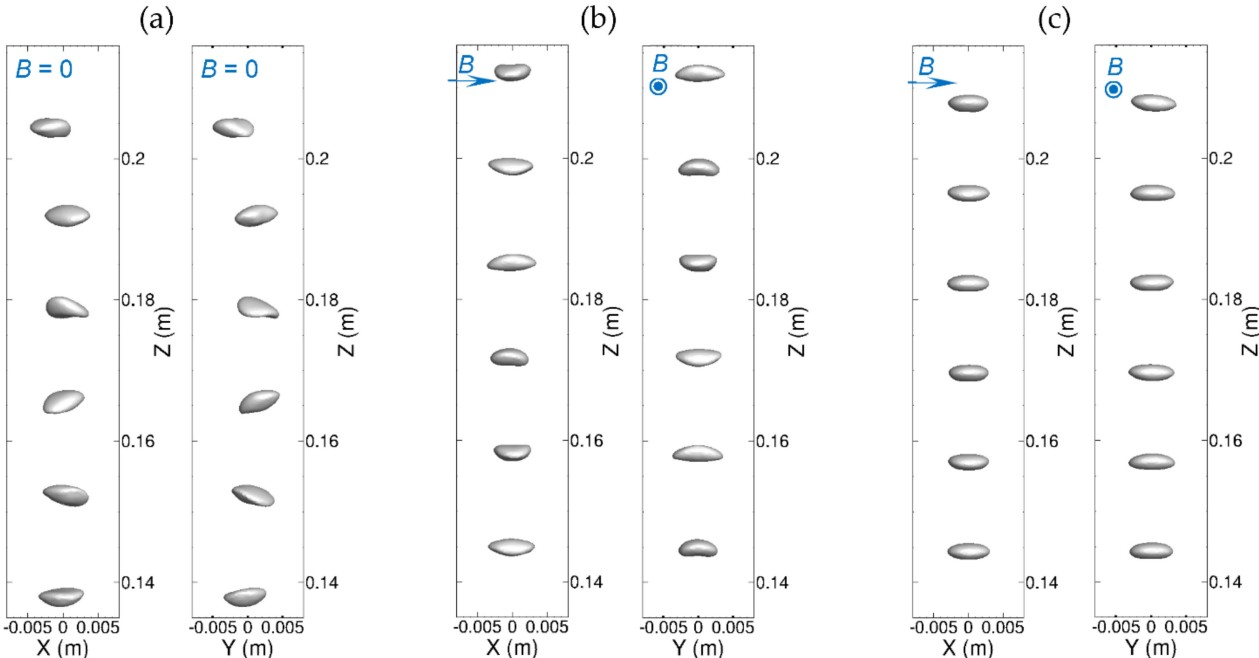

**Figure 15.** Instantaneous bubble shapes for (**a**) $B_0 = 0$ T, (**b**) $B_0 = 0.14$ T and (**c**) $B_0 = 0.28$ T at time intervals of 0.05 s. The trajectory for $B_0 = 0$ is not strictly vertical. The bubble shapes for $B_0 = 0.14$ T and 0.28 T change but remain laterally symmetric in the X–Z and Y–Z planes.

### 4.2. Wake Field

Referring again to Figure 13b, the terminal rise velocity is seen to decrease with an increase in the magnetic field strength for $B_0 \geq 0.28$ T. Here, we propose an explanation for this and discuss the mechanisms involved based on our computed results. The distributions of the fluid velocity relative to that of the rising bubble for $Eo = 2.44$ are shown in Figure 16.

The relative velocity is non-dimensional according to $u_{rel}{}^* = \frac{u - u_T}{|u_T|}$, where $u$ is the velocity vector with respect to the space-fixed coordinate system, and $u_T$ is the terminal rise velocity vector of the bubble. The blue-colored region below the bubble in Figure 16, i.e., for which $|u_{rel}{}^*| < 1$, represents the wake. The wake is asymmetric and non-periodic for $B_0 = 0$, but it becomes progressively more symmetric and displays more periodic behavior for $B_0 = 0.14$ T and $B_0 = 0.28$ T. From this observation, one can conclude that the wake field is stabilized by the action of the Lorentz force. The velocity field for $B_0 \geq 0.56$ T in the $y = 0$ plane (Figure 16 bottom row), i.e., parallel to the magnetic field lines, indicates that a considerably broader region below the bubble now constitutes the wake, and this becomes stronger with an increase in the magnetic field strength. In order to better understand the structure of the wake, three-dimensional views of the iso-surface $|u_{rel}{}^*| = 0.9$ are displayed in Figure 17. The value $|u_{rel}{}^*| = 0.9$ was selected because the iso-surface clearly visualizes the principal feature of the wake field. A long wake field broken up into small eddies is observed for $B_0 = 0$. The length of the wake behind the bubble is progressively shortened with increases in $B_0$, up to 0.28 T, but the iso-surface for $B_0 = 0.28$ T still features turbulence in the wake. However, the iso-surface turns out to be smooth, i.e., featuring a wake-like Stokes flow when $B_0$ reaches 0.56 T. The wake for $B_0 = 0.56$ T is not axisymmetric and is slightly elongated in the $x$-direction, but this broadens considerably for the cases of $B_0 \geq 1.12$ T.

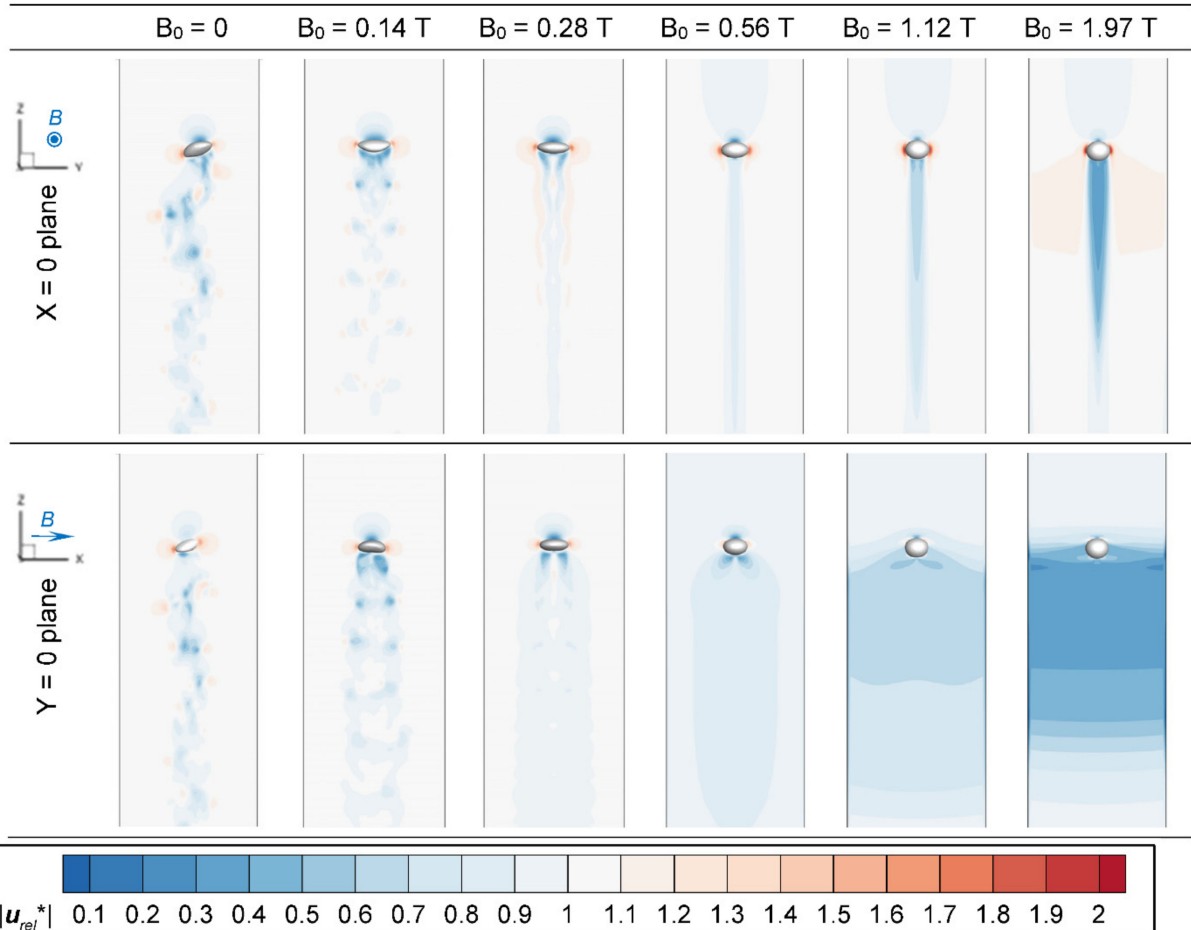

**Figure 16.** Distribution of the non-dimensional relative velocity $|u_{rel}{}^*|$ on the $x = 0$ plane (**top row**) and on the $y = 0$ plane (**bottom row**). The wake field is stabilized by the action of the Lorentz force and shows different features between planes $x = 0$ and $y = 0$ for higher $B_0$ ($\geq 0.56$ T).

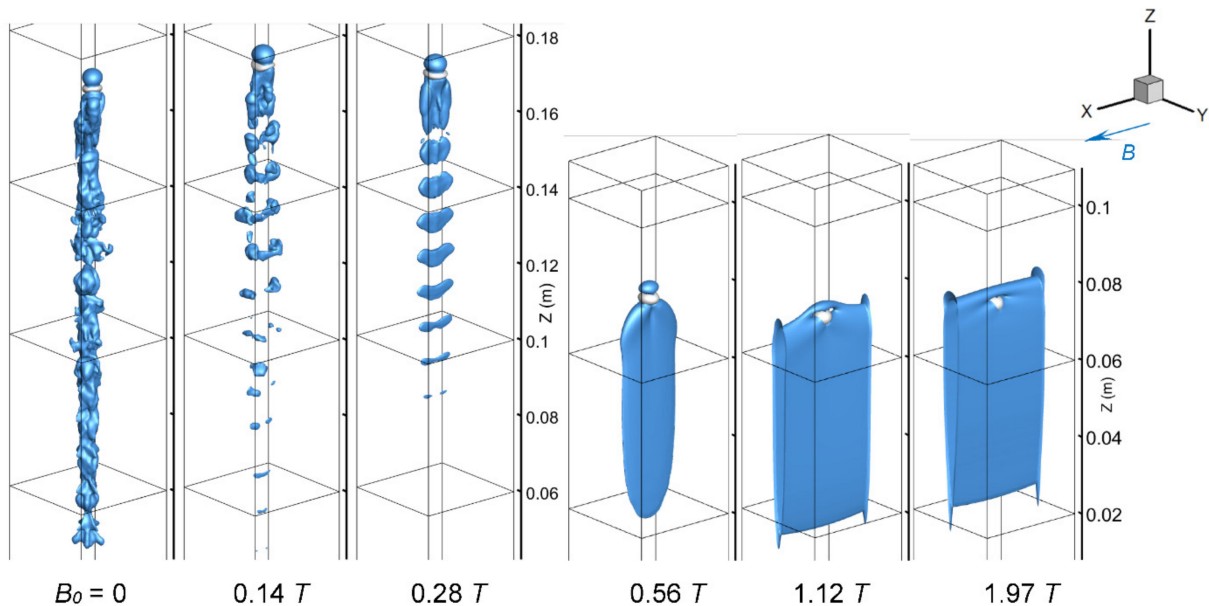

$B_0 = 0$      0.14 $T$      0.28 $T$      0.56 $T$      1.12 $T$      1.97 $T$

**Figure 17.** Three-dimensional image of the iso-surface $|u_{rel}{}^*| = 0.9$. The wake is apparently broader in the *x*-direction for higher $B_0$ ($\geq$1.12 T) (see Video S1).

In order to observe eddies in the wake, the iso-surface of the *z*-component of the non-dimensional vorticity $\omega_z^* = \omega_z \frac{d}{u_T} = \pm 0.457$ is depicted in Figure 18. The vortices, which are shed from the bubble, persist a long distance into the wake for the case of $B_0 = 0$, but they diffuse away with increases in the magnetic field. The vortex shedding is not observed for higher magnetic field intensities when $B_0 \geq 0.56$ T.

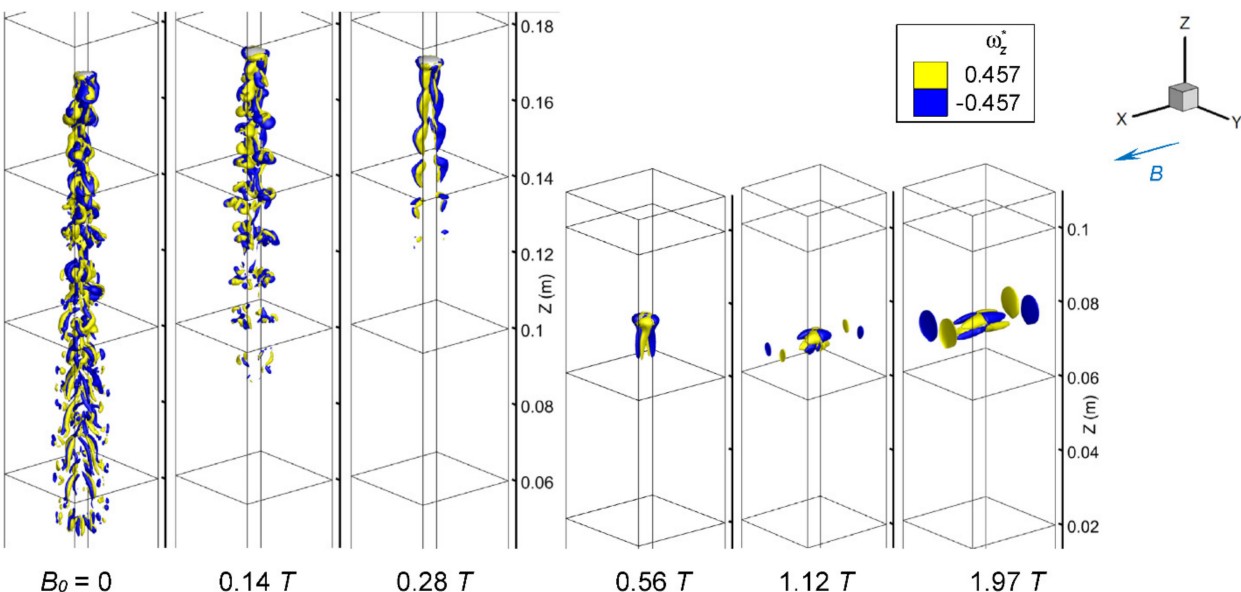

$B_0 = 0$      0.14 $T$      0.28 $T$      0.56 $T$      1.12 $T$      1.97 $T$

**Figure 18.** Iso-surfaces of non-dimensionalized vorticity $\omega_z^* = 0.457$ (yellow) and $\omega_z^* = -0.457$ (blue). The vortices, which are shed from the bubble, persist a long distance for the case of $B_0 = 0$, but they diffuse away with increases in the magnetic field (see Video S2).

It should be recalled that the peak in the terminal rise velocity appears at $B_0 = 0.14$, as displayed in Figure 13b. Based on the results shown in Figures 16–18, the increase in the rise velocity from $B_0 = 0$ to 0.14 T is considered to be caused by the shortening of the wake, i.e., the suppression of eddies in the downstream direction. The results also clarify that the

decrease in the rise velocity for higher magnetic field intensities when $B_0 \geq 0.56$ is caused by the growth of the wake. Rigorously speaking, the wake for $B_0 = 0.28$ T is apparently shorter than that for $B_0 = 0.14$ T, as shown in Figure 17, but the rise velocity for $B_0 = 0.28$ T is slightly faster than that for $B_0 = 0.14$ T, as seen in Figure 13b. These mechanisms, i.e., (i) the suppression of eddies in the wake for lower magnetic field intensities and (ii) the growth of the wake in a higher magnetic field intensity range, are investigated hereafter. Nonetheless, what we can learn from the current results is that the strength of the wake exhibits nonlinear behavior with increases in magnetic field intensity, and the weaker wake seems to result in a higher rise velocity.

### 4.3. Trajectory of Rising Bubbles

Since the rise velocities for the cases of $B_0 \leq 0.28$ T are unstable, as seen in Figure 13a, one can expect unstable trajectories to be one of the causes. In addition, the magnetic field applied to the simulations are anisotropic, i.e., only in the *x*-direction, and it is interesting to investigate the influence of the magnetic field on the bubble trajectory more closely. The trajectories of the centroids of the bubbles are displayed in Figure 19 for different values of $B_0$. Note that the trajectories for cases with $B_0 \geq 0.56$ T are rectilinear. The *x*- and *y*-coordinates of the centroid were calculated in the same way as for the *z*-coordinate, which is defined in Equation (13). The fluctuations in the trajectories are particularly noticeable in the region of $Z/d > 20$. The fluctuations observed for $B_0 = 0$ (red lines in Figure 19) are already seen to have been greatly suppressed for $B_0 = 0.14$ T. Comparing the trajectories in the X–Z and Y–Z planes, i.e., the blue lines in Figure 19a,b, one can notice that the trajectory in the Y–Z plane is more rectilinear than that in the X–Z plane. This is because the Lorentz force suppresses fluctuations in the *x*-direction more strongly than in the *y*-direction because the force vector operates in a perpendicular direction to the magnetic field, which, in this case, is the *y*-direction.

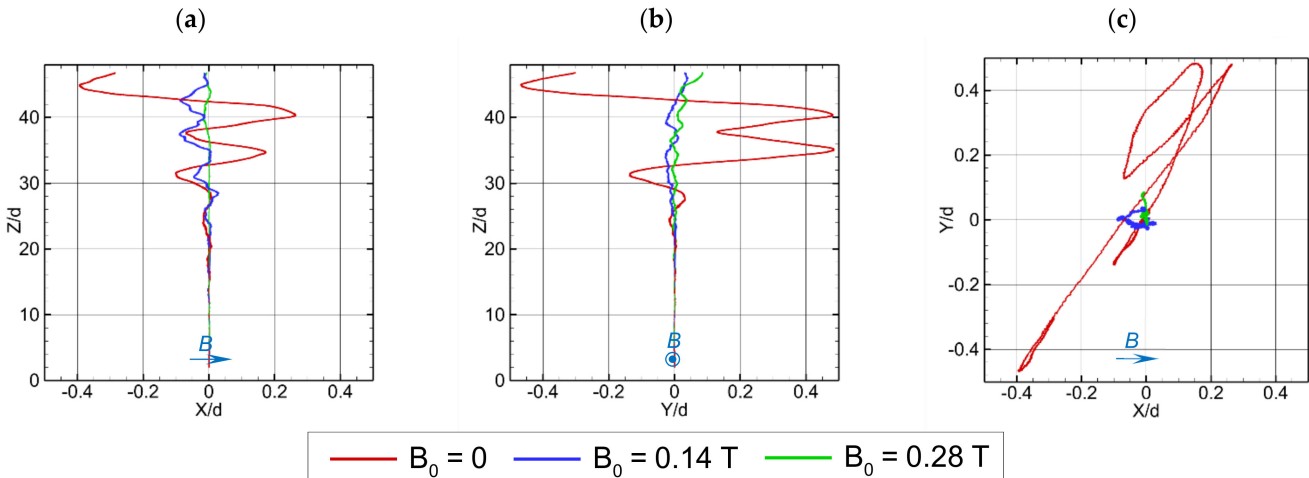

**Figure 19.** Trajectories of the rising bubbles for different magnetic field intensities in the (**a**) X–Z, (**b**) Y–Z and (**c**) X–Y planes. The trajectories become straighter with increases in $B_0$. Comparing the blue lines in (**a**,**b**), the trajectory in the Y–Z plane is more rectilinear than that in the X–Z plane because the Lorentz force suppresses fluctuations in the *x*-direction more strongly than in the *y*-direction.

The reason why a horizontal magnetic field makes bubble trajectory rectilinear is investigated in the next section through the visualization of the Lorentz force.

The corresponding result for $B_0 = 0$ can be characterized from its location in the bubble-regime diagram, which was first proposed by Grace, et al. [47], as shown in Figure 20. The Reynolds number, based on the computed terminal rise velocity of a bubble of diameter $d = 4.57$ mm ($Eo = 2.44$), for $B_0 = 0$, is $Re = 4200$. The computed unstable bubble shape seen in Figure 15a is consistent with the "wobbling" regime in the Grace diagram, as depicted in

Figure 20. Note that the bubble rise velocities in liquid metal available in the literature [8,45] lie in the wobbling regime, as indicated by a shaded region in Figure 20.

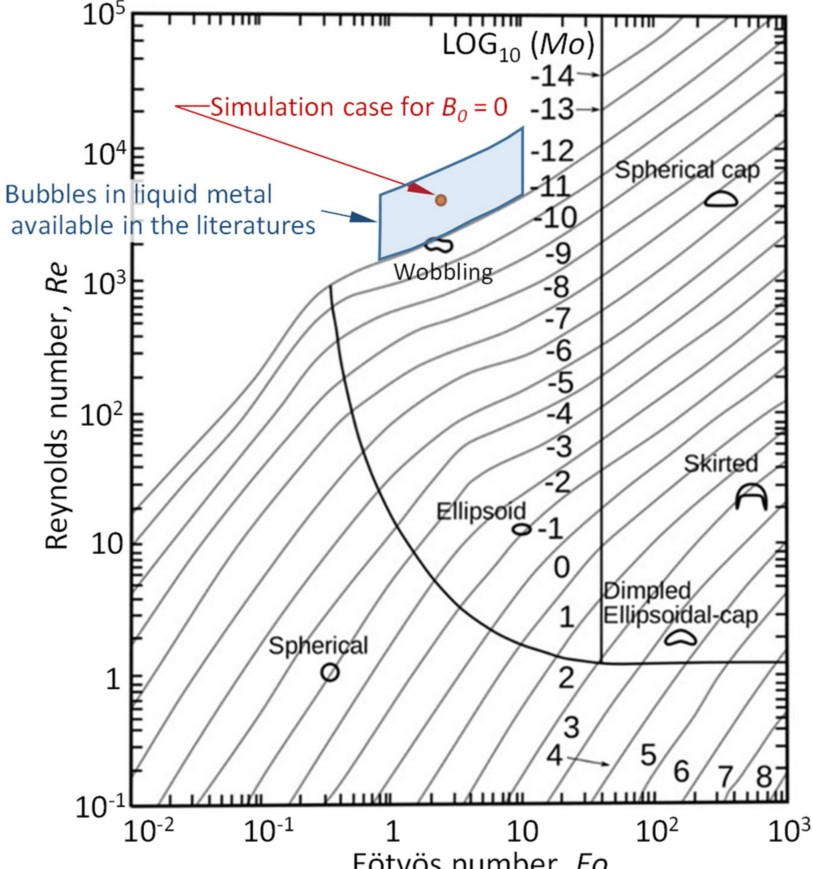

**Figure 20.** Bubble characterization for $B_0 = 0$ in the diagram of Grace, et al. [47]. The simulation result for $B_0 = 0$ corresponds to the wobbling regime.

*4.4. Lorentz Force*

In this section, we investigate the influence of the Lorentz force on the suppression of the bubble trajectory and the wake field.

4.4.1. Lorentz Force for Lower Magnetic Field

First, we investigate the result in the case of a lower magnetic field intensity. The distribution of the Lorentz force vectors for the case of $B_0 = 0.14$ T is shown in Figure 21. In the $x = 0$ plane, the Lorentz force vectors around the top half of the bubble are in a direction normal to the interface, pointing from the liquid phase to the gas phase, and those around the bottom half, although also normal to the interface, point from the gas phase to the liquid phase. The Lorentz force distribution in the $y = 0$ plane shows that the forces at the side of the bubble point upward, and this result qualitatively agrees with the simulations of Jin, et al. [17] and Zhang, et al. [18]. The Lorentz force being directed upward, which is colored green in Figure 21b, is the primary cause of the large wake, elongated in the $x$-direction.

The distributions of the flow velocity and the Lorentz force vectors in the $x = 0$ plane are compared in Figure 22. An interesting feature is that the Lorentz force vectors point in the opposite direction to the flow velocity vectors in the liquid, which indicates that the flow was decelerated by the action of the force, and it explains why the stability of the bubble trajectory increases and its wake decreases as a result of the presence of the magnetic field.

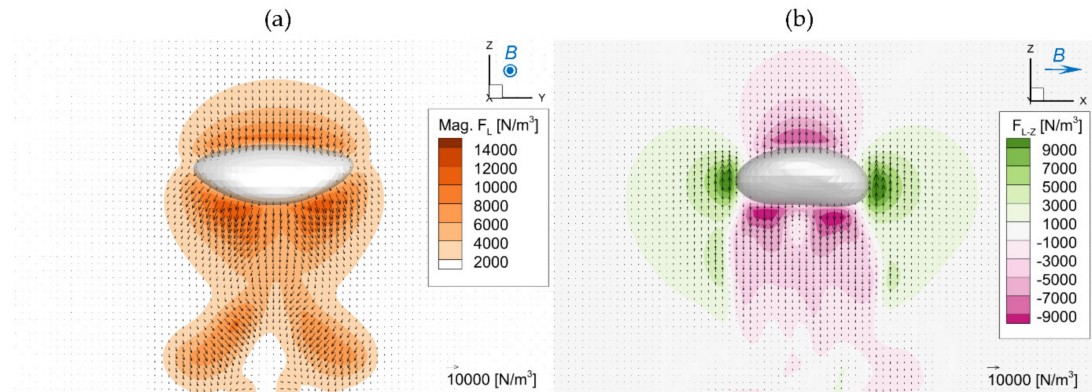

**Figure 21.** Distribution of Lorentz force vectors for $B_0 = 0.14$ T: (**a**) in the $x = 0$ plane, including the color contour of the absolute magnitude of the force, and (**b**) in the $y = 0$ plane, with the color contour here representing the $z$-component of the force. Note that the $x$-component of Lorentz force is zero in the y = 0 plane.

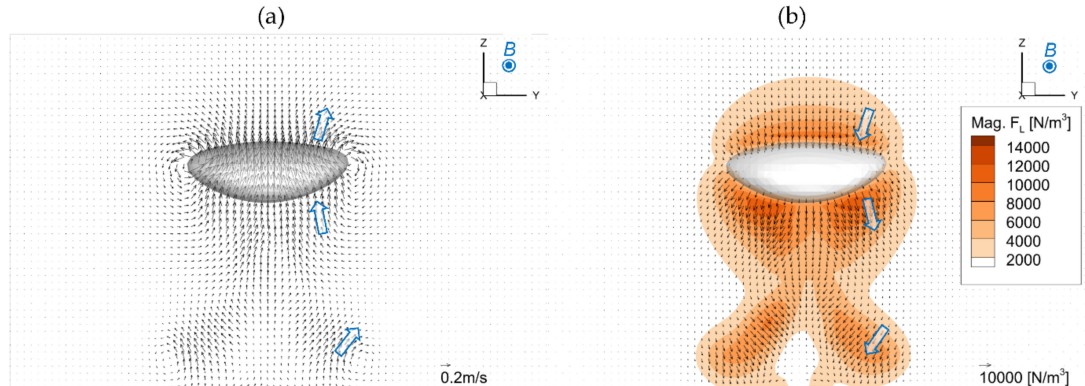

**Figure 22.** Comparison of (**a**) flow velocity vector distribution and (**b**) Lorentz force distribution for $B_0 = 0.14$ $T$ in the $x = 0$ plane. Lorentz force vectors point in the opposite direction to the flow velocity vectors. The bold arrows represent the main feature of the flow velocity (**a**) or the force (**b**).

However, such a mechanism is not reproduced in the $y = 0$ plane, as shown in Figure 23. The $x$-component of the Lorentz force is zero here, since:

$$F_L = j \times B = (j_x, j_y, j_z)^T \times (B_0, 0, 0)^T = (0, B_0 j_z, -B_0 j_y), \tag{16}$$

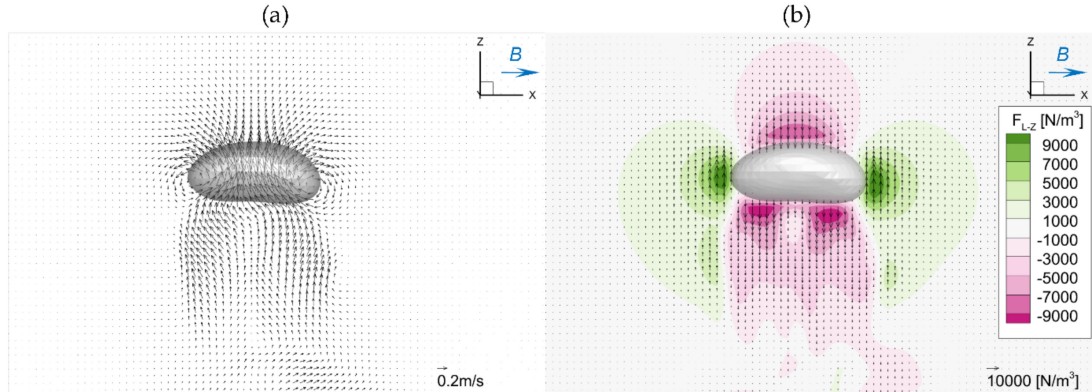

**Figure 23.** Comparison of (**a**) velocity vector distribution and (**b**) Lorentz force distribution for $B_0 = 0.14$ T in the $y = 0$ plane.

This indicates that the stability of the bubble in the $x$-direction is not directly influenced by the Lorentz force. Indeed, the trajectory for $B_0 = 0.14$ T in the Y–Z plane (blue line in Figure 19b) is more rectilinear than that in the X–Z plane (blue line in Figure 19a), which indicates that the magnetic field primarily stabilizes the bubble motion in the $y$-direction only in the case of the present orientation of the magnetic field in the $x$-direction.

Next, we clarify how this Lorentz force distribution was generated around the bubble. As described in Section 2.1, the Lorentz force is defined as: $\boldsymbol{F}_L = \boldsymbol{j} \times \boldsymbol{B}$. In the simulations reported here, the magnetic field is set to the static value of $\boldsymbol{B} = (B_0, 0, 0)$, and the Lorentz force can therefore be simplified to:

$$\boldsymbol{F}_L = \left(0,\ B_0 j_z,\ -B_0 j_y\right), \tag{17}$$

where $j_y$ and $j_z$ are the $y$- and $z$-components of the electrical current density, respectively. The distribution of the electrical current density vector around the bubble is visualized in Figure 24a. The larger electric current vectors are directed from the left side of the bubble to the right side. The electric current density $\boldsymbol{j}$ may be decomposed into that which is due to the magnetic field, $\boldsymbol{j}_B$, and into that which derives from the electric potential, $\boldsymbol{j}_\phi$, according to:

$$\boldsymbol{j} = \boldsymbol{j}_B + \boldsymbol{j}_\phi,\ \boldsymbol{j}_B = \sigma \boldsymbol{u} \times \boldsymbol{B},\ \boldsymbol{j}_\phi = -\sigma \nabla \phi, \tag{18}$$

which are visualized in Figure 24a,b. Note that $\boldsymbol{j}_B$ is calculated simply from the flow velocity field $\vec{u}$, since $\boldsymbol{B}$ is assumed to be uniform and static, and $\boldsymbol{j}_\phi$ is obtained in such a way that the divergence-free condition is strictly satisfied, i.e., $\nabla \cdot \boldsymbol{j} = 0$.

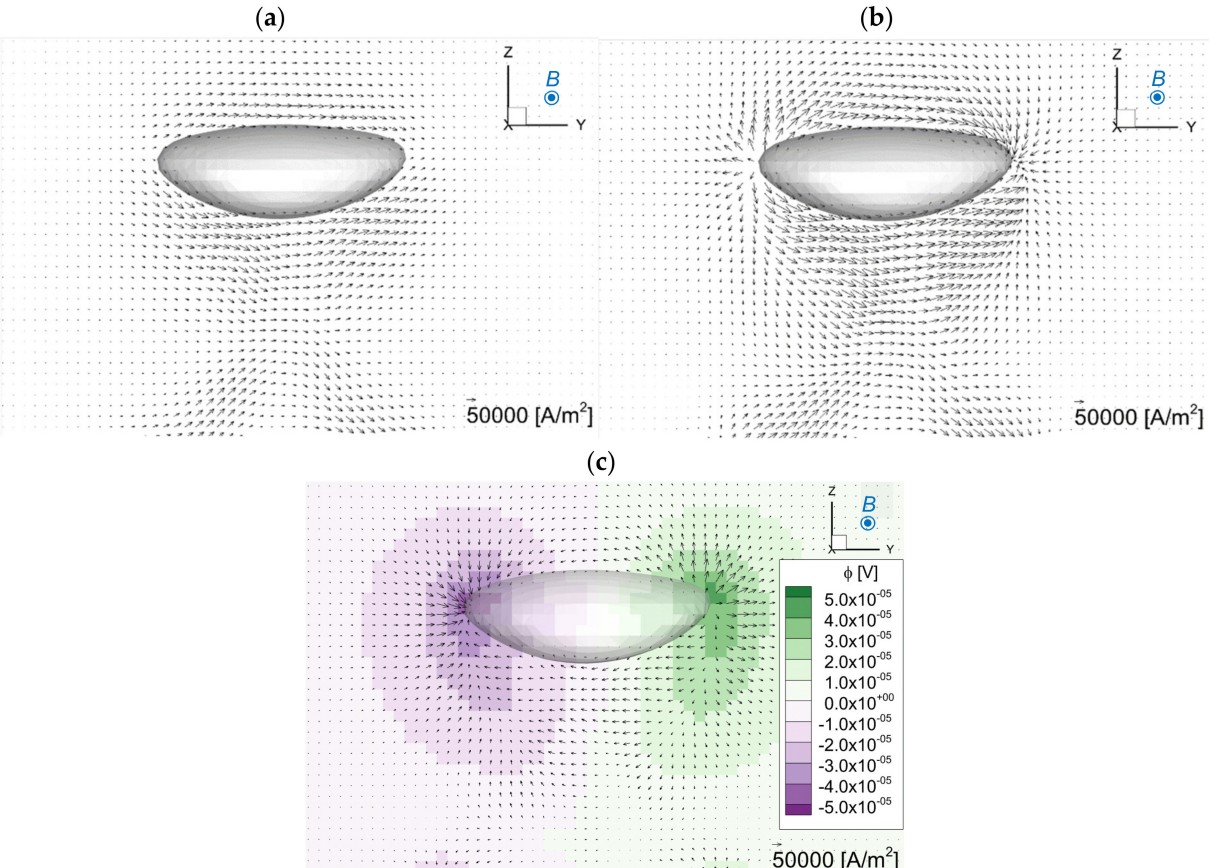

**Figure 24.** Distributions of the electric current vector and its components in the $x = 0$ plane for $B_0 = 0.14$ T: (**a**) $\boldsymbol{j}$, (**b**) $\boldsymbol{j}_B$ and (**c**) $\boldsymbol{j}_\phi$ together with the electric potential $\phi$ as the color contour. The distribution of $\boldsymbol{j}_B$ generally resembles that of $\boldsymbol{j}$.

In Figure 24, the distribution of $j_B$ generally resembles that of $j$ in total, i.e., the electric current is directed from the left side of the bubble to the right side. Moreover, the distribution of $j_\phi$ exhibits the opposite behavior, especially around the side of the bubble. Note that $j_\phi$ is directed toward the lower values of $\phi$ from the higher values, since $j_\phi = -\sigma \nabla \phi$, as seen in Figure 24c. Note that the $x$-component of $j_B$ is zero, since $j_B = \sigma u \times B$ and $B = (B_0, 0, 0)$.

We now try to clarify the reason why the directions of the flow velocity and the Lorentz force in the $x = 0$ plane are almost opposite to each other, as seen in Figure 22. The computed electric currents exhibit the similarity between $j$ and $j_B$, except at the sides of the bubble. Based on this similarity, one can approximate the total Lorentz force as follows:

$$j \approx j_B = \sigma u \times B = \sigma B_0(0,\ w,\ -v), \tag{19}$$

where $v$ and $w$ are the velocity components in the $y$- and $z$-directions, respectively. Substituting Equation (19) into Equation (17), we obtain

$$F_L \approx \sigma B_0^2(0,\ -v,\ -w) = -\sigma B_0^2 u, \tag{20}$$

which indicates that the Lorentz force vector in the $x = 0$ plane is in the opposite direction to the flow velocity. Consequently, the flow velocity in this plane is decelerated as a consequence of the action of the Lorentz force, and the bubble motion and wake development in this plane are suppressed, as previously noted in regard to Figure 19.

### 4.4.2. Lorentz Force for Higher Magnetic Field

Following the investigation of the influence of the strength of the Lorentz force for lower magnetic field intensities (Section 4.4.1), the force for higher magnetic field intensities is analyzed in this section. First, we focus on the Lorentz force in the $y = 0$ plane. The distribution of the force for different magnetic field strengths is shown in Figure 25a–c, where the distribution of the $z$-component of the force $F_{L\text{-}Z}$ in the $y = 0$ plane is depicted. As mentioned before, the $x$-component of the Lorentz force is zero in this plane, and the force in the region painted green acts toward the upper direction. As the applied magnetic field intensity increases, the intensity of $F_{L\text{-}Z}$ increases, and the distribution of the force also changes. The green region is elongated in the lateral direction in the case of a large magnetic field intensity (Figure 25c). To evaluate the influence of the intensity quantitatively, the maximum value of $F_{L\text{-}Z}$ in the $y = 0$ plane is as shown in Figure 25d. As can be seen, the maximum value changes steeply in the lower magnetic field.

To investigate the interaction between the Lorentz force and the wake behind the rising bubbles, a three-dimensional view of the $z$-component of the Lorentz force, together with the iso-surface $|u_{rel}{}^*| = 0.9$, is visualized in Figure 26. The Lorentz force, acting upward, i.e., the green region, induces the wake, which is broadened in the $x$-direction. The distribution of the Lorentz force is symmetric in the $x$-direction, and the bubble rise is rectilinear. This result clearly explains that the wake for higher magnetic field intensities is induced by the Lorentz force acting upward in the $y = 0$ plane (green region).

Next, we investigate the Lorentz force in the $x = 0$ plane. It should be recalled that the Lorentz force points in the opposite direction for the flow velocity in a lower magnetic field intensity, as seen in Figure 22, which decelerates the flow velocity, shortens the wake and results in a higher bubble rise velocity as a consequence. Thus, it would be interesting to know if this mechanism works even in a higher magnetic field. The distribution of the Lorentz force vector and velocity vector in the $x = 0$ plane is illustrated in Figure 27. In the vicinity of the bubble surface, the Lorentz force vectors and velocity vectors point in the opposite direction, which is a similar situation to that illustrated in Figure 22. However, in the region beneath the bubble, the velocity vectors point upward (red arrow in Figure 27b), whereas a large intensity of the Lorentz force vectors cannot be observed in the corresponding region in Figure 27a. Consequently, the strong wake beneath the bubble

is not weakened by the Lorentz force, though eddies surrounding the bubble surface may be eliminated by the Lorentz force, i.e., vortex shedding does not take place.

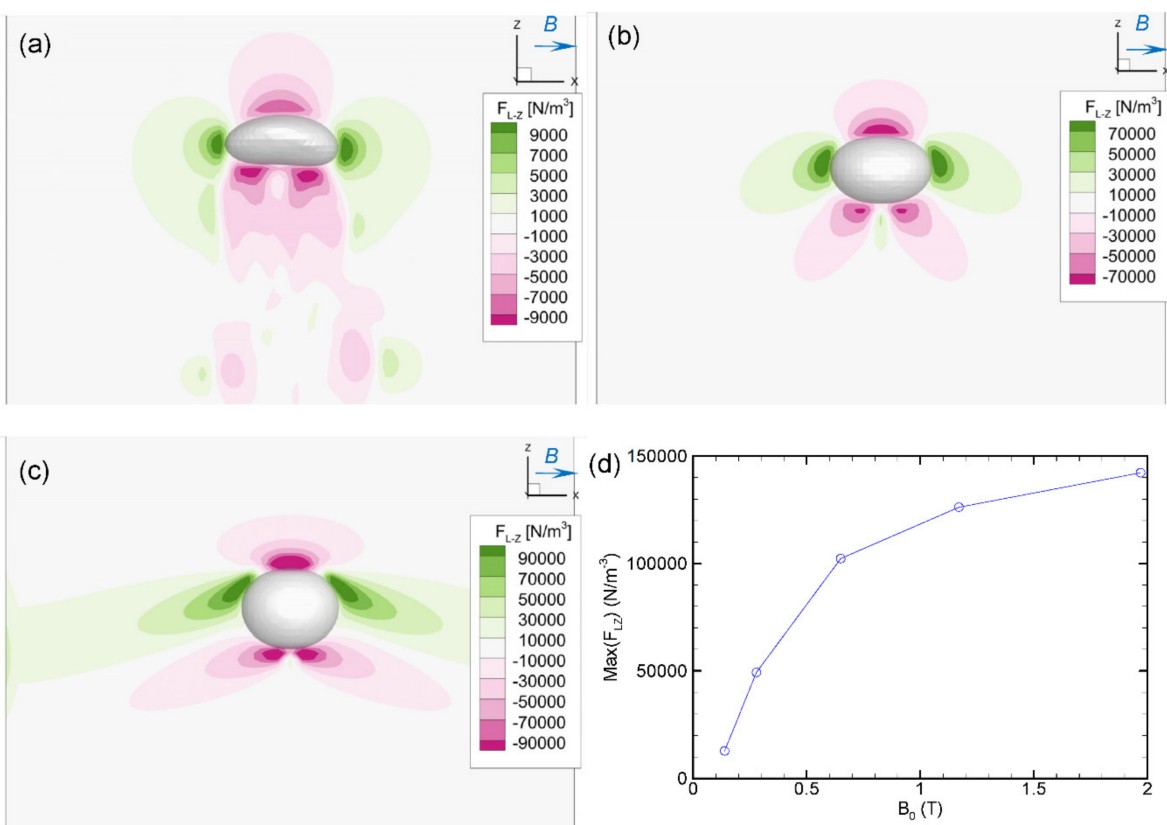

**Figure 25.** Distribution of Lorentz force in the $y = 0$ plane for (**a**) $B_0 = 0.14$ T, (**b**) $B_0 = 0.56$ T, (**c**) $B_0 = 1.12$ T and (**d**) the maximum value of the $z$-component of the Lorentz force in the $y = 0$ plane. The Lorentz force vectors are not depicted in (**a**–**c**) since the x-component of the vector is zero, as shown in Figure 23b.

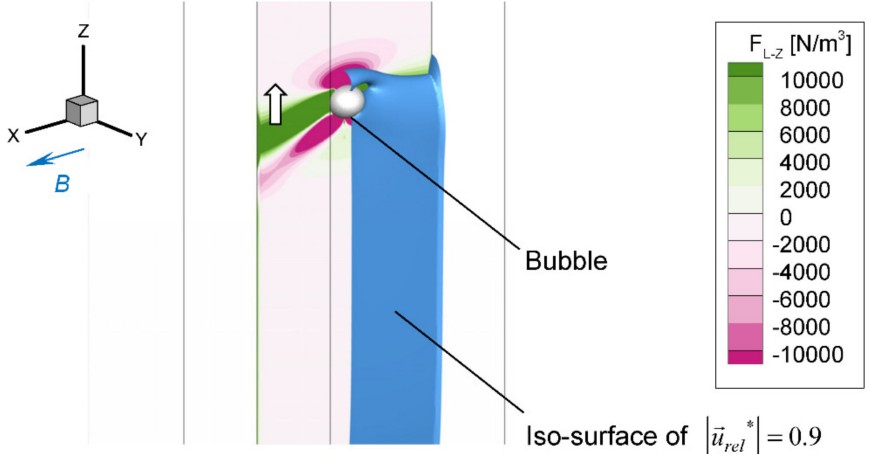

**Figure 26.** Wake induced by the Lorentz force for $B_0 = 1.12$ T showing the iso-surface of the magnitude of the relative velocity $|u_{rel}{}^*| = 0.9$, drawn only in the half domain for the sake of clarity. An upward-directed force is identified by the color green. The Lorentz force, acting upward (the green region), induces the wake, which is broadened in the $x$-direction.

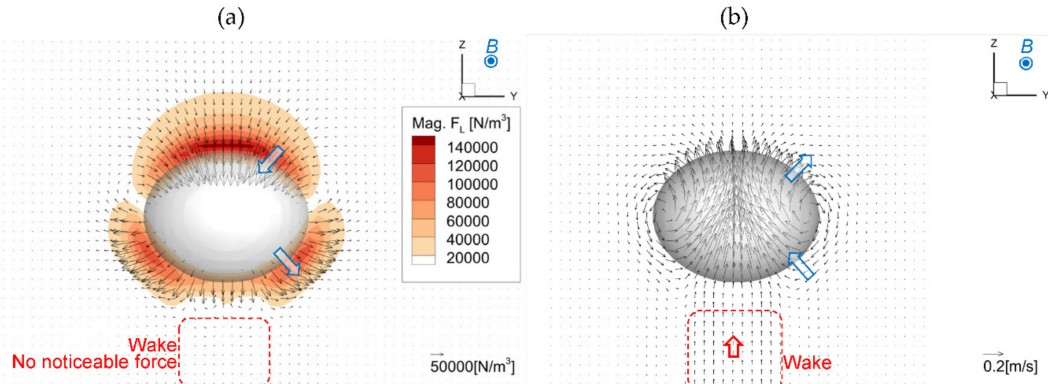

**Figure 27.** Distribution of (**a**) the Lorentz force vectors and (**b**) velocity vectors in the $x = 0$ plane for $B_0 = 1.12$ T. In the wake, noticeable Lorentz force was not observed, i.e., the Lorentz force does not decelerate the flow velocity. The bold arrows (blue and red) represent the main feature of the flow velocity, and the frame drawn with red, dashed line indicates the wake.

In order to understand the Lorentz force distribution in the higher magnetic field region, the electric current and its components are visualized in Figure 28. In this case, $j$ shows a similar distribution to $j_B$ ($j \approx j_B$), and the Lorentz force is directed opposite to the flow velocity, as derived in Equation (20). In general, the electric currents $j$ and $j_B$ point from the left side of the bubble to the right side, but the discrepancy can be found beneath the bubble. The electric current $j_B$, in the wake region, is directed from the left to the right side almost homogeneously, but it is cancelled out by $j_\phi$. As a consequence, the Lorentz force in the wake region is very small due to the minuteness of the total electric current $j$.

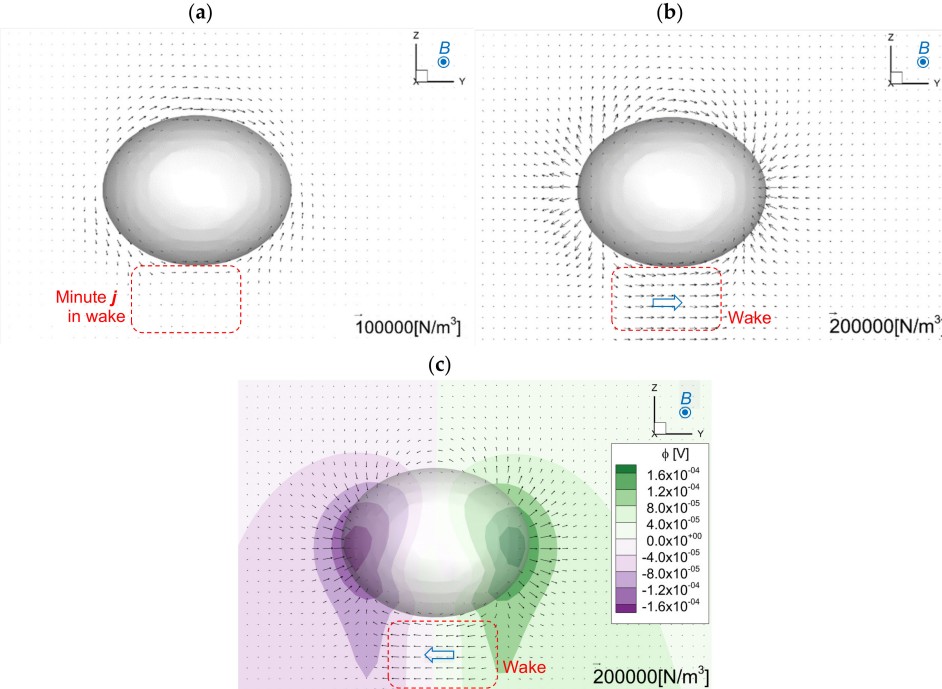

**Figure 28.** Distributions of the electric current vectors and its components in the $x = 0$ plane for $B_0 = 1.12$ T: (**a**) $j$, (**b**) $j_B$ and (**c**) $j_\phi$ together with the electric potential $\phi$ as the color contour. The bold blue arrows represent the main feature of the electric current in the wake field indicated by the red frame. The distribution of $j_B$ does not resemble that of $j$, especially in the wake region, which is different from the condition in lower magnetic field intensity, as shown in Figure 24.

## 5. Conclusions

In this paper, we simulated a single bubble rising in a stagnant liquid metal under the influence of a horizontal magnetic field for the purposes of understanding the mechanism of the nonlinear behavior of the rise velocity as a function of the intensity of the applied magnetic field.

First, we developed an MHD code by implementing the electric potential method and the computation of the Lorentz force into PSI-BOIL, a CFD code originally developed to simulate two-phase flows with phase changes. To support the model's development, verification and validation exercises were performed. The first refers to the flow of mercury in a square duct driven by an applied pressure drop and in the presence of a static lateral magnetic field for the purpose of verifying the modeling of the MHD equations in single-phase flow. The computed flow rate, for this configuration, shows very good agreement with the analytical solution of Shercliff and with the measurements of Hartmann and Lazarus.

The validation case involves a single bubble rising in a liquid metal subject to an applied horizontal magnetic field. The range of the investigated Eötvös numbers was $1.12 \leq Eo \leq 3.67$, with the Hartmann number in the range of $0 \leq Ha \leq 425$. The computed terminal rise velocity under zero magnetic field conditions was seen to be slightly higher than that derived from the well-known Tomiyama correlation. Grid dependency studies were undertaken, and they demonstrated near-second-order accuracy in space for the bubble rise velocity.

Based on our simulation results, we further investigated and proposed an explanation of the observed nonlinear behavior of the terminal rise velocity of a bubble as a function of the intensity of the applied magnetic field, specifically the mechanism for (i) the observed increase in the terminal velocity of the bubble with increases in the magnetic field strength in the lower magnetic intensity region ($B_0 \leq 0.28$ T) and (ii) the decrease in the terminal velocity in the higher intensity region ($B_0 \geq 0.28$ T). In the subsequent analysis of the computed results, the electric current $\boldsymbol{j}$ was decomposed into two components: one due to the electric potential, $\boldsymbol{j}_\phi = -\sigma \nabla \phi$ and the other resulting from the magnetic field $\boldsymbol{j}_B = \sigma \boldsymbol{u} \times \boldsymbol{B}$. The distinction proved to be significant in understanding the behavior of the rising bubble.

The nonlinear behavior of the rise velocity as a function of the intensity of the applied magnetic field is the result of two factors: (a) decreases in drag due to the suppression of vortex shedding, and (b) increases in drag due to the wake induced by the Lorentz force. The suppression of vortex shedding (a) is caused by the Lorentz force, directed almost in an opposite direction to the flow velocity primarily in the plane perpendicular to the magnetic field since the force decelerates the flow velocity. The opposite directions of the Lorentz force and the flow velocity are attributed to the similarity of the distribution of $\boldsymbol{j}$ to that of $\boldsymbol{j}_B$, i.e., $\boldsymbol{F}_L \approx -\sigma B_0^2 \boldsymbol{u}$, in the $x = 0$ plane. Vortex shedding disappears for magnetic field strengths with a range of $0.28$ T $< B_0$. This mechanism $\boldsymbol{F}_L \approx -\sigma B_0^2 \boldsymbol{u}$ is not observed in the wake field for higher magnetic field intensities because the distribution of $\boldsymbol{j}$ differs from $\boldsymbol{j}_B$, and thus the wake is not suppressed by the Lorentz force. The increase in drag due to the wake (b) is induced by the Lorentz force being directed upward on the sides of the bubble, acting in the plane parallel to the magnetic field. This Lorentz force becomes larger with increases in the magnetic field intensity and forms a large wake behind it, which is broadened in the direction of the magnetic field. This results in a decrease in the rise velocity. We anticipate that the same tendency, i.e., the increases/decreases in terminal velocity and the straightened trajectory of the bubble, can be accurately measured in an experiment using X-ray or neutron tomography by our colleagues at the Paul Scherrer Institute in the near future, and in this paper, we laid the foundations of an associated numerical simulation.

**Supplementary Materials:** The following supporting information can be downloaded at: https://www.mdpi.com/xxx/s1. Video S1: Time evolution of the wake field (Figure 17). Video S2: Time evolution of the vorticity field (Figure 18).

**Author Contributions:** Conceptualization, M.C. and Y.S.; methodology, M.C. and Y.S.; software, M.C. and Y.S.; validation, M.C. and Y.S.; formal analysis, M.C. and Y.S.; investigation, M.C. and Y.S.; writing—original draft preparation, M.C. and Y.S.; writing—review and editing, M.C. and Y.S.; visualization, M.C. and Y.S.; supervision, Y.S.; project administration, Y.S. All authors have read and agreed to the published version of the manuscript.

**Funding:** This research received no external funding.

**Data Availability Statement:** Detailed data may be obtained upon request from the corresponding author.

**Acknowledgments:** This work was partially supported by a grant from the Swiss National Supercomputing Centre (CSCS) under project ID psi05. The authors would like to thank Brian Smith for valuable technical discussions, advice and English corrections to the manuscript. We also thank Konstantin Mikityuk for the discussion of the results.

**Conflicts of Interest:** The authors declare no conflict of interest.

## Nomenclature

Abbreviations
| | |
|---|---|
| CFD | computational fluid dynamics |
| HMF | horizontal magnetic field |
| MHD | magneto-hydro-dynamics |
| UDV | ultrasound Doppler velocimetry |
| VMF | vertical magnetic field |
| VOF | volume of fluid |

Physical quantities
| | |
|---|---|
| $a$ | half channel width (m) |
| $B, \boldsymbol{B}$ | magnetic flux density and flux density vector (T) |
| $B_0$ | applied static magnetic flux density (T) |
| $d$ | bubble diameter based on bubble volume (m) |
| $\boldsymbol{F}$ | body force (N/m$^3$) |
| $\boldsymbol{g}$ | gravitational acceleration vector (m/s$^2$) |
| $\boldsymbol{j}$ | electrical current density vector (A/m$^2$) |
| $k$ | applied pressure drop for channel flow (N/m$^3$) |
| $LZ$ | computational domain height (m) |
| $t$ | time (s) |
| $\boldsymbol{u}$ | flow velocity vector (m/s) |
| $u_T$ | terminal rise velocity (m/s) |
| $u, v, w$ | velocity components in the x-, y- and z-directions, respectively (m/s) |
| $V$ | volume of computational cell (m$^3$) |
| $\alpha$ | volume fraction of liquid (–) |
| $\gamma$ | coefficient of surface tension (N/m) |
| $\eta$ | magnetic diffusivity (m$^2$/s) |
| $\mu$ | dynamic viscosity coefficient (Pa.s) |
| $\rho$ | density (kg/m$^3$) |
| $\sigma$ | electrical conductivity (1/Ωm) |
| $\phi$ | electric potential (V) |
| $\omega$ | vorticity (1/s) |

Subscripts
| | |
|---|---|
| $g$ | gas |
| $L$ | Lorentz force |
| $l$ | liquid |
| $rel$ | relative |

Superscript

| | |
|---|---|
| $n$ | time step |

Dimensionless numbers

| | |
|---|---|
| $C_d$ | drag coefficient, $C_d = \dfrac{(\rho_l - \rho_g)gV}{\frac{1}{2}\rho_l u_T^2 \frac{\pi}{4}d^2} = \dfrac{4d(\rho_l - \rho_g)g}{3\rho_l u_T^2}$ |
| $Eo$ | Eötvös number, $Eo = \dfrac{(\rho_l - \rho_g)gd^2}{\gamma}$ |
| $Ha$ | Hartmann number for a rising bubble, $Ha = dB\sqrt{\dfrac{\sigma_l}{\mu_l}}$ |
| $Ha_{channel}$ | Hartmann number for channel flow, $Ha_{channel} = aB\sqrt{\dfrac{\sigma_l}{\mu_l}}$ |
| $Mo$ | Morton number, $Mo = \dfrac{g\mu_l^4(\rho_l - \rho_g)}{\rho_l^2 \gamma^3}$ |
| $N$ | Stuart number, $N = \dfrac{B^2 d\sigma_l}{\rho_l u_T}$ |
| $Re$ | bubble Reynolds number, $Re = \dfrac{\rho_l u_T d}{\mu_l}$ |
| $Rm$ | magnetic Reynolds number, $Rm = \dfrac{u_T d}{\eta_l}$ |

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
