# Peer review of "Magnetohydrodynamics Simulation of the Nonlinear Behavior of Single Rising Bubbles in Liquid Metals in the Presence of a Horizontal Magnetic Field"

_fluids, doi:10.3390/fluids7110349_

Round 1

Reviewer 1 Report

The manuscript fluids-1931795 “Magnetohydrodynamics simulation of the nonlinear behavior of single rising bubbles in liquid metals in the presence of a horizontal magnetic field” by Marino Corrado and Yohei Sato discusses the topic of rising argon bubbles in GaInSn in a horizontal magnetic field. The authors adapt existing multiphase code PSI-BOIL by introducing a Lorentz force calculation and use the code to replicate experimental results presented by Wang. The authors claim to explain a previously poorly understood effect, namely the decrease of terminal bubble velocity in a higher magnetic field (B>~0.3). The manuscript is written and organized using grammatically correct English and is generally well-written and easily readable. The manuscript does a good job of introducing key aspects of bubble rise physics covered previously, and the remaining gaps. Although the range of bubble sizes investigated here are limited and some of the results are contrary to experimental findings, I believe the work warrants publication with some minor revisions. I detail my suggestions below.

General content revision:

·        I believe the fluids community would benefit from some comments on how bubble rise is affected when the size is very small (Eo<<1) and very large (Eo>>5). This is especially true in applications where the bubble size distribution can be wide.

·        The authors mention “contamination” of experimental bubbles as a hypothesis for lack of matching velocity trends and absolute values. What is meant by “contamination”? Is this a presence of gallium oxide? Lipids from handling? Other impurities in the bulk metal? Either way, some comments on this would be great since contamination of liquid metal free surfaces has implications on many free surface flow phenomena and making this connection would make the paper more interesting to a wider community.

·        I believe some of the results could be accompanied by supplemental videos or animations which would allow other investigators to compare their results and derive more information from the simulations. In particular, figures 8, 15, 17 and 18 would be more easily interpretable if one could track the time evolution of the isosurfaces.

Specific content revision:

·        In line 53, the Eotvos number is declared without defining it first. The authors should define how they use the number for researchers in adjacent fields

·        The x and y axis labels in Figure 4 should have prime markers (y’ not Y), since these are normalized

·        A higher DPI render for Figure 4 would be great, it is difficult to resolve the contour/vector in figure 4(c)

·        A gravity vector would be great in Figure 6 for those readers who will be skimming the paper figures

·        In line 236, the authors propose that Marangoni stress suppresses surface motion and thus increases the drag coefficient. The mechanism of increased drag coefficient is unclear. Please include either further explanation or references that further explain this effect.

·        The legend in Figure 11 interferes with the plotted lines. I would suggest reformatting this plot to avoid this.

·        In lines 482 and 93, the ratio N/Cd are used without defining how the drag coefficient Cd was calculated. Please define how the Stuart number and how the drag coefficient values were calculated (e.g. what was used as the characteristic length?)

·        In lines 549 – 554 the authors claim that the length of the wake is directly related to the terminal velocity. If this is truly a direct relationship, shouldn’t the terminal velocity be highest for B=0.28T? Figure 17 shows that the B=0.28T case creates an even shorter wake than B=0.14, and yet the terminal velocity is lower. The authors should clarify this discrepancy.

·        In figure 25, the vector field is missing. I would suggest adding it so that this figure can be compared directly to figure 23(b)

Author Response

Please find attached the document, which replies to the reviewer's comment.

Reviewer 2 Report

The authors presented an excellent paper on the nonlinear behavior of single rising bubbles in liquid metals in the presence of a horizontal magnetic field.

The paper is well prepared and can be accepted after minor revision:

The dimensions of the computational domain are to be justified.

The boundary conditions are to be expressed mathematically.

The authors considered 3D configuration; it will be interesting to present the 3D flow structure.

Have you used a moving mesh, to follow the shape modification of the bubble ? to be explained.

Author Response

Please find attached the file.
